# QuIP: 2-Bit Quantization of Large Language Models With Guarantees

**Jerry Chee**
Cornell University
jerrychee@cs.cornell.edu

**Yaohui Cai**
Cornell University
yc2632@cornell.edu

**Volodymyr Kuleshov**
Cornell University
kuleshov@cornell.edu

**Christopher De Sa**
Cornell University
cdesa@cs.cornell.edu

## Abstract

This work studies post-training parameter quantization in large language models (LLMs). We introduce quantization with incoherence processing (QuIP), a new method based on the insight that quantization benefits from *incoherent* weight and Hessian matrices, i.e., from the weights being even in magnitude and the directions in which it is important to round them accurately being unaligned with the coordinate axes. QuIP consists of two steps: (1) an adaptive rounding procedure minimizing a quadratic proxy objective; (2) efficient pre- and post-processing that ensures weight and Hessian incoherence via multiplication by random orthogonal matrices. We complement QuIP with the first theoretical analysis for an LLM-scale quantization algorithm, and show that our theory also applies to an existing method, OPTQ. Empirically, we find that our incoherence preprocessing improves several existing quantization algorithms and yields the first LLM quantization methods that produce viable results using only two bits per weight. Our code can be found at https://github.com/Cornell-RelaxML/QuIP.

## 1 Introduction

Large language models (LLMs) have enabled advances in text generation, few-shot learning, reasoning, protein sequence modeling, and other tasks [2, 30, 35]. The massive size of these models—often reaching into hundreds of billions of parameters—requires sophisticated deployment methods and motivates research into efficient inference algorithms.

This work studies the post-training quantization of LLM parameters as a way to improve their runtime efficiency [4, 8, 22, 31, 33, 34]. Our key insight is that quantization can be most effective when weight and proxy Hessian matrices are *incoherent*—that the weights themselves are even in magnitude, and the directions in which it is important to have good rounding accuracy are not too large in any one coordinate. Intuitively, incoherence can be thought of as a principled form of outlier reduction, which makes it easier to adaptively round the weights to a finite set of compressed values. We use this intuition to develop theoretically sound two-bit quantization algorithms that scale to LLM-sized models.

Specifically, we introduce quantization with incoherence processing (QuIP), a new method motivated by the above insight. QuIP consists of two steps: (1) an adaptive rounding [20] procedure, which minimizes a quadratic proxy objective $\ell(\hat{W}) = \mathrm{tr}((\hat{W} - W)H(\hat{W} - W)^T)$ of the error between the original weights $W$ and the quantized weights $\hat{W}$ using an estimate of the Hessian $H$; (2) efficient pre- and post- processing that ensures that the weight and Hessian matrices are incoherent by multiplying

37th Conference on Neural Information Processing Systems (NeurIPS 2023).

them by a Kronecker product of random orthogonal matrices. We denote "incoherence processing" as both the pre- and post- processing steps of our procedure. Incoherence processing can be viewed as a form of outlier suppression across the weights and the activation space.

We complement our method with a theoretical analysis—the first for a quantization algorithm that scales to LLM-sized models—which analyzes the role of incoherence and shows that our quantization procedure is optimal within a general class of rounding methods. Interestingly, we find that QuIP without incoherence processing yields a more efficient implementation of an earlier algorithm, OPTQ [8]; our paper thus also provides the first theoretical analysis for that method.

Empirically, we find that incoherence processing greatly improves the quantization of large models, especially at higher compression rates, and yields the first LLM quantization method that produces viable results using only two bits per weight. For large LLM sizes (>2B parameters), we observe small gaps between 2-bit and 4-bit compression that further decrease with model size, hinting at the feasibility of accurate 2-bit inference in LLMs.

**Contributions.** In summary, this paper makes the following contributions: (1) we propose QuIP, a quantization method based on the insight that model parameters should ideally be incoherent; (2) we provide a theoretical analysis for a broad class of adaptive rounding methods that encompass QuIP and OPTQ; (3) we demonstrate that QuIP makes two-bit LLM compression viable for the first time.

## 2    Related Work

**Adaptive rounding.** Nagel et al. [20] are the first to motivate the "adaptive rounding" proxy objective (Eq. (1)) in a principled way. There are many quantization methods which quantize by optimizing this proxy objective [5, 6, 9, 12, 14, 20, 32]. Many require further retraining which can be expensive, and are not evaluated on the current largest open LLMs (OPT [35], BLOOM [30]). Lybrand and Saab [15] propose a greedy per-neuron quantization procedure that is similar to ours, except they do not consider arbitrary linear functions of the error correction. Their work bounds the proxy objective, albeit on the first layer only.

**Post training quantization in large models.** There is a growing body of work on PTQ in LLMs such as OPT and BLOOM. The size of these models make it difficult to apply previously developed methods. The majority of these methods make quantization easier by somehow reducing the range of weights or activations, but still use nearest rounding. SmoothQuant [31] rescales between activations and weights to remove outliers from the activations and make quantization overall easier. ZeroQuant [33] proposes a per-layer knowledge distillation method. LLM.int8() [4] decompose matrix multiplications into a majority of 8 bit and a minority of 16 bit operations. LUT-GEMM [22] designs kernels to accelerate quantized matrix multiplications. RPTQ [34] reorders activations and quantizes them in groups, reducing the impact of range differences between channels.

**OPTQ (Formerly known as GPTQ).** OPTQ [8] is based on OBQ [7], and proposes a novel rounding method that can work on the largest OPT and BLOOM models. The method works iteratively over the weight columns in a fixed order: (1) quantize with nearest rounding and compute the error, (2) update the remaining weights with a scaled error, and (3) repeat.

**Other quantization methods.** There are other quantization procedures which do not round based on the proxy objective of [20], or are not designed for the largest language models [10, 11, 13, 19, 28, 29].

## 3    Quantization With Incoherence Processing: Adaptive Rounding Step

This section introduces quantization with incoherence processing (QuIP), a new method consisting of: (1) an adaptive rounding step; (2) efficient pre- and post-processing that ensures weight and Hessian incoherence. We define and analyze step (1) in this section; the next section focuses on step (2).

Following existing state-of-the-art post-training quantization methods, we round weights per-layer by minimizing the "adaptive rounding" proxy objective, as in Nagel et al. [20],

$$\ell(\hat{W}) = \mathbf{E}_x\left[\left\|(\hat{W} - W)x\right\|^2\right] = \mathrm{tr}\left((\hat{W} - W)H(\hat{W} - W)^T\right). \tag{1}$$

Here, $W \in \mathbb{R}^{m \times n}$ is the original weight matrix for a given linear layer, $\hat{W} \in \mathbb{R}^{m \times n}$ are the quantized weights, $x \in \mathbb{R}^n$ is an input vector drawn uniformly at random from a calibration set, and $H$ is the

second moment matrix of these vectors, interpreted as a proxy Hessian. Crucially, this formulation lets the quantization be run in parallel across neurons, which is tractable for large language models [8]. For simplicity, we will focus in this section on rounding to the integers; subsequent sections will extend the analysis to finite grids.

## 3.1 LDLQ: An Optimal Adaptive Rounding Method

Our strategy is to define a family of adaptive rounding methods for optimizing objective (1) and then define LDLQ, the optimal method within that class. Our defined methods iteratively perform the following update for $k = 1, 2, ..., n$:

$$\hat{W}_k = \mathcal{Q}(W_k + (W_{1:(k-1)} - \hat{W}_{1:(k-1)})a_k),$$

where $W_k$ denotes the $k$-th column, $W_{1:(k-1)}$ denotes the first $k-1$ columns, the subroutine $\mathcal{Q}$ denotes either nearest rounding or standard unbiased rounding to the integers (which rounds up or down such that $\mathbf{E}\left[\mathcal{Q}(z)\right] = z$), and $a_k \in \mathbb{R}^{k-1}$ is some sequence of vectors. This scheme rounds columns one at a time; at each step, it adds a "correction" term that is a linear function of the residual from the rounding we have done so far. The final $\hat{W}$ satisfies the following matrix equation:

$$\hat{W} = \mathcal{Q}(W + (W - \hat{W})U), \tag{2}$$

where $U$ is a strictly upper-triangular matrix whose columns are the vectors $a_k$ and $\mathcal{Q}$ acts elementwise. Because $U$ is upper-triangular, $\hat{W}_k$ only depends on $\hat{W}_{1:(k-1)}$.

If we let $\eta = \mathcal{Q}(W + (W - \hat{W})U) - (W + (W - \hat{W})U)$ denote the quantization error of $\mathcal{Q}$, we find that $\hat{W} - W = \eta(U + I)^{-1}$ and we can rewrite objective (1) as

$$\text{tr}((\hat{W} - W)H(\hat{W} - W)^T) = \text{tr}(\eta(U + I)^{-1}H(U + I)^{-T}\eta^T). \tag{3}$$

**The LDLQ Method**  How should we specify $U$, the linear feedback from the quantization error of preceding columns in (2)? Equation 3 provides an answer. If we choose $U \leftarrow \dot{U}$ such that the LDL decomposition of $H$ is

$$H = (\dot{U} + I)D(\dot{U} + I)^T, \tag{4}$$

where $D$ is a (non-negative) diagonal matrix and $\dot{U}$ is upper unit triangular, then the terms $(U + I)$ in Eq. (3) cancel. We denote as LDLQ the rounding procedure in Eq. (2) with $U \leftarrow \dot{U}$ as the LDL assignment from Eq. (4). We will now see that the LDL assignment of $U$ is in fact optimal.

## 3.2 Deriving the Optimality of the LDLQ Adaptive Rounding Procedure

In order to reason about optimality, we consider weights which are worst and average-case for the proxy loss. Let $\mathcal{A}$ denote a rounding method, and let $\mathcal{A}(W, H)$ be the resulting quantized weights. Define the *worst-case* ($\mathcal{L}_{\text{worst}}$) and *average* ($\mathcal{L}_{\text{avg}}$) proxy losses with respect to the input weights as

$$\mathcal{L}_{\text{worst}}(\mathcal{A}, H) = \sup_{W \in \mathbb{R}^{m \times n}} \mathbf{E}\left[\text{tr}\left((\mathcal{A}(W, H) - W)H(\mathcal{A}(W, H) - W)^T\right)\right] \tag{5}$$

$$\mathcal{L}_{\text{avg}}(\mathcal{A}, H) = \mathbf{E}_{W \sim \text{Unif}[0,1]^{m \times n}}\left[\text{tr}\left((\mathcal{A}(W, H) - W)H(\mathcal{A}(W, H) - W)^T\right)\right]. \tag{6}$$

**Theorem 1.** LDLQ *is worst and average-case optimal amongst rounding methods which specify the linear feedback $U$ as a function of $H$ (not of $W$), and when rounding to the integers. That is, for all rounding methods $\mathcal{A}$ in the class described by Eq. (2), for all positive semi-definite $H$, and for $\mathcal{Q}$ as either nearest or stochastic rounding,*

$$\tfrac{m}{4}\text{tr}(D) = \mathcal{L}_{\text{worst}}(\text{LDLQ}, H) \leq \mathcal{L}_{\text{worst}}(\mathcal{A}, H) \quad \text{and} \quad \tfrac{m}{c}\text{tr}(D) = \mathcal{L}_{\text{avg}}(\text{LDLQ}, H) \leq \mathcal{L}_{\text{avg}}(\mathcal{A}, H),$$

*where $D$ is the matrix from the LDL decomposition of $H$, and $c = 12$ for nearest, $c = 6$ for stochastic.*

**Remarks.** The number of rows being quantized is $m$, and each quantization method operates across the $n$ entries of each row. For all rounding methods described by Eq. (2), and for all positive semi-definite $H$, $\mathcal{Q}$ as nearest rounding achieves the same worst-case proxy loss as stochastic rounding, but achieves better average proxy loss.

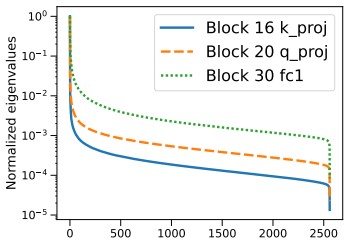 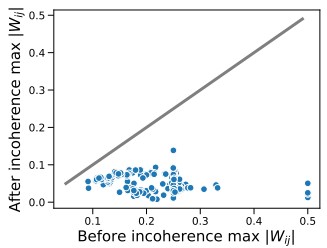 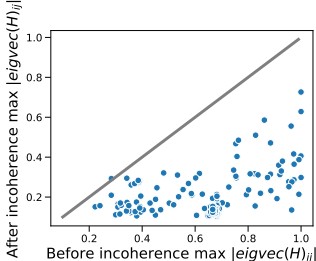

Figure 1: $\mathrm{eig}(H)$ from OPT-2.7b.

Figure 2: Max $|W_{ij}|$ before and after incoherence processing on OPT-2.7b.

Figure 3: Max $|Q_{ij}|$ before and after incoherence processing, with $Q$ the eigenvectors of $H$ on OPT-2.7b.

Moving beyond a generic algorithm $\mathcal{A}$ within our framework, we consider the common baselines of nearest and stochastic rounding. These methods are represented within our framework by choosing the appropriate $\mathcal{Q}$ subroutine, and setting all entries of the linear feedback to zero.

For these baseline methods, their optimality gap to LDLQ is governed by $\mathrm{tr}(D)$ vs. $\mathrm{tr}(H)$. For any non-diagonal $\tilde{H} \succeq 0$, LDLQ achieves strictly lower worst and average-case proxy loss because $\mathrm{tr}(D) < \mathrm{tr}(\tilde{H})$. Let $\mathcal{B} = \{\mathsf{Near}, \mathsf{Stoch}\}$. Then, $\mathcal{L}_{\mathrm{worst}}(\mathsf{LDLQ}, \tilde{H}) < \mathcal{L}_{\mathrm{worst}}(\mathsf{Stoch}, \tilde{H})$ and $\mathcal{L}_{\mathrm{avg}}(\mathsf{LDLQ}, \tilde{H}) < \mathcal{L}_{\mathrm{avg}}(\mathcal{B}, \tilde{H})$. Across OPT models 125m to 2.7b, $\mathrm{tr}(D)/\mathrm{tr}(H) \leq 0.65$—empirically verifying that the gap is not insignificant. See Supplement C for full details.

### 3.3 Incoherence: Optimality with a Spectral Bound

Theorem 1 gives exact expressions for the proxy loss, albeit with $\mathrm{tr}(D)$, which can be difficult to reason about. In Figure 1, we empirically observe that $H$ is approximately low-rank: we visualize the spectrum of several randomly chosen $H$ from OPT-2.7b, and observe that the spectrum decays rapidly. In fact, across all layers of OPT-125m to 2.7b models, a vast majority of $H$ matrices have fewer than a quarter of eigenvalues $> 1\%$ of the max eigenvalue; see Supplement C for full details. Given this observation about the low rank of $H$, can we bound the behavior of LDLQ, and thus $\mathrm{tr}(D)$, using the spectrum of $H$?

We do this building on a variant of the incoherence assumption that is specialized to our case [3, 24].

**Definition 1.** *We say a symmetric Hessian matrix $H \in \mathbb{R}^{n \times n}$ is $\mu$-incoherent if it has an eigendecomposition $H = Q\Lambda Q^T$ such that for all $i$ and $j$, $|Q_{ij}| = |e_i^T Q e_j| \leq \mu/\sqrt{n}$. By extension, we say a weight matrix $W \in \mathbb{R}^{m \times n}$ is $\mu$-incoherent if all $i$ and $j$, $|W_{ij}| = |e_i^T W e_j| \leq \mu \|W\|_F /\sqrt{mn}$.*

Note that "most" $n \times n$ matrices are incoherent with $\mu = \mathcal{O}(\sqrt{\log n}) = \tilde{\mathcal{O}}(1)$ because a random orthogonal matrix has entries with squared-magnitudes that concentrate around their mean of $1/n$. Incoherence in $W$ can be viewed as a form of outlier reduction: a small bound on the magnitude of its entries means that we do not need to scale it as much to make it fit in the finite range of representable low-precision numbers. Figures 2 and 3 plot the max absolute weight and hessian eigenvector entries before and after our incoherence processing, on all layers in OPT-2.7b. A line with slope=1 is drawn for reference. We see that $W$ and $H$ are more incoherrent after our incoherence processing is applied. Making $H$ incoherent is less intuitive, but its utility is motivated by the following lemma.

**Lemma 2.** *Let $H \in \mathbb{R}^{n \times n}$ be a $\mu$-incoherent positive semi-definite symmetric matrix and let $H = (\dot{U} + I)D(\dot{U} + I)^T$ be its LDL Cholesky decomposition, where $\dot{U}$ is a strictly upper triangular matrix and $D$ is a (non-negative) diagonal matrix. Then,*

$$\mathrm{tr}(D) \leq \frac{\mu^2}{n} \mathrm{tr}\left(H^{1/2}\right)^2.$$

To the best of our knowledge, this is a novel result using incoherence to obtain a bound on $\mathrm{tr}(D)$ that depends only on the spectrum of $H$. To help interpret this result, we derive explicit proxy losses for plain nearest and stochastic rounding, which we will then compare to what LDLQ gets via Lemma 2.

**Algorithm 1** QuIP - Incoherence Pre-Processing

**Require:** $b \in \mathbb{N}$, $H \in \mathbb{R}^{n \times n}$ SPD, original $W \in \mathbb{R}^{m \times n}$, $\rho \in \mathbb{R}_+$, $\alpha \in [0, 1]$
1: **seeded sample** random two-factor orthogonal matrices $U \in \mathbb{R}^{m \times m}$ and $V \in \mathbb{R}^{n \times n}$
2: $H = H + \alpha * \mathrm{mean}(\mathrm{diag}(H))I$             ▷ from OPTQ
3: $\tilde{D} \leftarrow \sqrt[4]{\mathrm{diag}(H)/\mathrm{diag}(W^T W)}$          ▷ $\sqrt[4]{\phantom{x}}$ applies element-wise
4: $W \leftarrow W\tilde{D}; \quad H \leftarrow \tilde{D}^{-1}H\tilde{D}^{-1}$          ▷ diagonal rescaling
5: $W \leftarrow UWV^T; \quad H \leftarrow VHV^T$            ▷ incoherence
6: $s \leftarrow \rho\|W\|_F/\sqrt{mn}; \quad W \leftarrow \frac{1}{2}(\frac{1}{s}W + 1)$    ▷ reduced quantization range due to incoherency
7: $W \leftarrow \mathrm{clamp}(W * (2^b - 1), 0, 2^b - 1)$        ▷ rescale $W$ to lie within $[0, 2^b - 1]$
8: **return** $W, H, s, \tilde{D}$

---

**Algorithm 2** QuIP - Incoherence Post-Processing

**Require:** $b \in \mathbb{N}$, $H \in \mathbb{R}^{n \times n}$ SPD, quantized $W \in [0, 2^b - 1]^{m \times n}$, $s \in \mathbb{R}$ & $\tilde{D} \in \mathbb{R}^{n \times n}$ (Alg 1)
1: **seeded sample** random two-factor orthogonal matrices $U \in \mathbb{R}^{m \times m}$ and $V \in \mathbb{R}^{n \times n}$
2: $W \leftarrow s * \big((W/(2^b - 1)) * 2 - 1\big)$
3: $W \leftarrow U^T WV; \quad H \leftarrow V^T HV$           ▷ revert incoherence
4: **return** $W \leftarrow W\tilde{D}^{-1}$             ▷ revert diagonal rescaling

---

**Lemma 3.** *Let $H$ be symmetric positive definite. In the worst case stochastic rounding achieves $\mathcal{L}_{\mathrm{worst}}(\mathsf{Stoch}, H) = (m/4)\,\mathrm{tr}\,(H)$. In the average case nearest and stochastic rounding achieve $\mathcal{L}_{\mathrm{avg}}(\{\mathsf{Near}, \mathsf{Stoch}\}, H) = (m/c)\,\mathrm{tr}\,(H)$, where $c = 12$ for nearest, and $c = 6$ for stochastic.*

To interpret this result, consider $H$ rank-$k$ with $\mu^2 k < n$. By Cauchy-Schwarz, $\mathrm{tr}(H^{1/2})^2 \le k\,\mathrm{tr}\,(H)$. Combining Lemma 2 with the LDLQ proxy losses of Theorem 1 and comparing with Lemma 3,

$$\mathcal{L}_{\mathrm{worst}}(\mathsf{LDLQ}, H) \le \frac{m\mu^2}{4n}\,\mathrm{tr}\left(H^{1/2}\right)^2 \le \frac{m\mu^2 k}{4n}\,\mathrm{tr}\,(H) \le \frac{m}{4}\,\mathrm{tr}\,(H) = \mathcal{L}_{\mathrm{worst}}(\mathsf{Stoch}, H)$$

$$\mathcal{L}_{\mathrm{avg}}(\mathsf{LDLQ}, H) \le \frac{m\mu^2}{cn}\,\mathrm{tr}\left(H^{1/2}\right)^2 \le \frac{m\mu^2 k}{cn}\,\mathrm{tr}\,(H) \le \frac{m}{c}\,\mathrm{tr}\,(H) = \mathcal{L}_{\mathrm{avg}}(\mathcal{B}, H),$$

where $\mathcal{B} \in \{\mathsf{Near}, \mathsf{Stoch}\}$, and $c$ is as given in Theorem 1. This shows that for sufficiently low-rank $H$, LDLQ is asymptotically better than plain nearest and stochastic rounding by a factor of $\mu^2 k/n$.

**Without incoherence: no improvement with a spectral bound.** By assuming incoherence, we were able to show LDLQ gets an asymptotically better bound in terms of just the spectrum of $H$. We might ask: *was the incoherence assumption necessary to get this result?* The following theorem answers this question in the affirmative by showing that without incoherence, the best spectral bound for LDLQ cannot differentiate it from the nearest and stochastic rounding baselines.

**Theorem 4.** *Consider all $\tilde{H}$ with the same spectrum as $H$. For any positive semi-definite $H$, the following holds. On the worst-case loss* LDLQ *achieves the same error as stochastic rounding,*

$$\sup_{\tilde{H}\,s.t.\,\mathrm{eig}(\tilde{H})=\mathrm{eig}(H)} \mathcal{L}_{\mathrm{worst}}(\mathsf{LDLQ}, \tilde{H}) = \mathcal{L}_{\mathrm{worst}}(\mathsf{Stoch}, H) = \frac{m}{4}\,\mathrm{tr}\,(H).$$

*On the average-case loss* LDLQ *achieves the same error as the corresponding rounding routine. Let $\mathcal{B} = \{\mathsf{Near}, \mathsf{Stoch}\}$ and $c = 12$ for nearest, $c = 6$ for stochastic.*

$$\sup_{\tilde{H}\,s.t.\,\mathrm{eig}(\tilde{H})=\mathrm{eig}(H)} \mathcal{L}_{\mathrm{avg}}(\mathsf{LDLQ}^*, \tilde{H}) = \mathcal{L}_{\mathrm{avg}}(\mathcal{B}, H) = \frac{m}{c}\,\mathrm{tr}\,(H).$$

Note that the worst case for comparing LDLQ against these baselines occurs when $H$ is diagonal, see Theorem 1 and Lemma 3. Assuming incoherence as we do is a natural way to exclude such cases.

## 4    Quantization With Incoherence Processing: Incoherence Processing Step

Next, we leverage the above incoherence analysis to introduce *incoherence processing*, the second step of the QuIP algorithm. Our strategy will be to pre-process weight and Hessian matrices to ensure

the favorable incoherence properties outlined above. One straightforward way to make a symmetric matrix incoherent is to conjugate it by a uniform random orthogonal matrix: this will result in each of its eigenvectors being a random unit vector, whose entries will concentrate around magnitude $n^{-1/2}$.

Specifically, let $U \in \mathbb{R}^{m \times m}$ and $V \in \mathbb{R}^{n \times n}$ be two random orthogonal matrices. (Let's temporarily ignore how these matrices are generated, or how we would efficiently perform inference.) We ensure the weight and Hessian are incoherent with high probability through random orthogonal multiplications $\tilde{H} \leftarrow VHV^T$ and $\tilde{W} \leftarrow UWV^T$. Importantly, this transformation preserves the proxy quadratic form since $\mathrm{tr}(\tilde{W}\tilde{H}\tilde{W}^T) = \mathrm{tr}((UWV^T)(VHV^T)(VW^TU^T)) = \mathrm{tr}(WHW^T)$.

## 4.1 Incoherence via Efficient Orthogonal Multiplication

If all we wanted to do was to store or transmit the weights of the quantized neural network, the above procedure would introduce no overhead, since we can generate a random orthogonal matrix from a seed—making it essentially free to store. However, for running *inference* on a DNN, we need to multiply by the weight matrix $W$, and here the need to manifest and multiply by $n \times n$ random orthogonal matrices $U, V$ would be prohibitive.

To handle this, we propose to instead use a distribution over random orthogonal matrices for which multiplication is fast. Let $n = pq$ be a factorization of $n$ (where $p \approx q \approx \sqrt{n}$), and set $U = U_L \otimes U_R$ where $U_L$ is sampled uniformly from the $p \times p$ orthogonal matrices and $U_R$ is sampled uniformly from the $q \times q$ orthogonal matrices. Multiplication of a vector $x \in \mathbb{R}^n$ by the matrix $U$ can be accomplished by reshaping to a $p \times q$ matrix, multiplying on the left by $U_L$ and the right by $U_R^T$, and then reshaping back: this takes $O(n(p + q)) = o(n^2)$ operations. Using more than two factors in this way is also possible, but using two suffices to make this preprocessing asymptotically non-dominant.

**Lemma 5.** *Let $H$ be a positive semi-definite matrix on $\mathbb{R}^{n \times n}$ and $W$ a matrix on $\mathbb{R}^{m \times n}$, and suppose that $m = p_1 \cdot p_2 \cdots p_k$ and $n = q_1 \cdot q_2 \cdots q_k$. Let $U_1, U_2, \ldots, U_k, V_1, V_2, \ldots, V_k$ be independent random orthogonal matrices on $\mathbb{R}^{p_i \times p_i}$ and $\mathbb{R}^{q_i \times q_i}$ respectively. Set $U$ as the Kronecker product $U = U_1 \otimes U_2 \otimes \cdots \otimes U_k$ and $V$ as $V = V_1 \otimes V_2 \otimes \cdots \otimes V_k$ Then $VHV^T$ is $\mu_H$-incoherent with probability at least $1 - \delta$, and $UWV^T$ is $\mu_W$-incoherent with probability at least $1 - \delta$, where*

$$\mu_H = A^{k/2} \log\left(\frac{Ckn^2}{\delta}\right)^{k/2} = \tilde{\mathcal{O}}(1) \quad and \quad \mu_W = A^k \log\left(\frac{2Ckmn}{\delta}\right)^k = \tilde{\mathcal{O}}(1)$$

*for some global constants $A$ and $C$ independent of $n$ and $k$.*

**Remarks.** This lemma means that multiplying by a random matrix in this family suffices to make a matrix incoherent with parameter $\mu$ only poly-logarithmic in the matrix size. In our experiments we use $k = 2$ factors to construct the orthogonal matrices $U, V$.

## 4.2 Additional Heuristics

We outline QuIP pre-processing and post-processing in Algorithms 1 and 2, respectively. In line 5 of Algorithm 1, we apply the aforementioned fast orthogonal multiplication procedure to ensure $W$ and $H$ are incoherent. We also randomly permute entries at the fast matrix multiplication step to prevent any correlation between attention heads from worsening performance. We introduce a number of additional heuristic improvements that further improve performance.

**Incoherence-Based Heuristics.** Line 4 diagonally rescales $W$ and $H$ to minimize $\ell(\hat{W}) \approx \mathrm{tr}(H)\|W\|_F^2$, effectively trading off the spectrum of these matrices to find a minimum. Motivated by the incoherence of $W$, Line 6 computes the quantization range depending on the spectrum $\|W\|_F$, instead of the typical $\max_{i,j}|W_{ij}|$. Our full QuIP procedure is described in Algorithm 3, which contains calls to the pre- and post-processing sub-steps in Algorithms 1 and 2.

**Greedy local search.** Our basic procedure yields a good initial guess with error guarantees. We can further lower the proxy loss by running coordinate descent after LDLQ (but before post-processing), updating the weights in the same order as in the initial pass. See Supplement B for full details.

---

**Algorithm 3** QuIP: Quantization with Incoherence Processing

---

**Require:** $b \in \mathbb{N}$, $H \in \mathbb{R}^{n \times n}$ SPD, $W \in \mathbb{R}^{m \times n}$, $\mathcal{Q} \in \{\text{Near}, \text{Stoch}\}$, $\rho \in \mathbb{R}_+$, $\alpha \in [0, 1]$

1: $\hat{W}, H, s, \tilde{D} \leftarrow \text{Alg } 1(b, H, W, \rho, \alpha)$           ▷ QuIP Incoherence Pre-Procesing

2: $H = (\dot{U} + I)D(\dot{U} + I)^{-1}$                ▷ LDL decomposition

3: **for** $k \in \{1, \dots, n\}$ **do** $\hat{W}_k \leftarrow \text{clamp}(\mathcal{Q}(W_k + (W - \hat{W})\dot{U}_k), 0, 2^b - 1)$     ▷ LDLQ

4: **return** $\hat{W} \leftarrow \text{Alg } 2(b, H, \hat{W}, s, \tilde{D})$            ▷ QuIP Incoherence Post-Processing

---

## 5 Extensions and Further Analyses

### 5.1 OPTQ is a Special Case of LDLQ

We prove a novel theoretical insight: QuIP without incoherence processing (i.e., LDLQ) is equivalent to a more efficient version of the OPTQ algorithm. That is, OPTQ falls under our class of adaptive rounding procedures with linear feedback, and is within-class optimal.

**Theorem 6.** *OTPQ [8] falls within the class of adaptive rounding procedures with linear feedback as described by Eq. (2), and is equivalent to LDLQ in Section 3.*

**Remarks.** To the best of our knowledge, this equivalence yields the first theoretical analysis of OPTQ. Even though the two methods are equivalent, LDLQ is more efficient. OPTQ's implementation requires a matrix inversion of $H$, and two Cholesky decompositions. Our implementation of LDLQ performs no matrix inversion, and only one Cholesky decomposition.

**Empirical Verification.** The quantized outputs of the OPTQ implementation [8] are shown to be exactly identical to the outputs of our LDLQ implementation. Synthetic random data was used, with $W \sim \text{Unif}[0, 1]^{1000 \times 1000}$. Full details can be found in Supplement C.

### 5.2 A Bound for Rounding to a Finite Grid

In Section 3, we saw that LDLQ (equivalently, OPTQ) is optimal for minimizing the adaptive rounding objective. However, this analysis assumed rounding to the integers. In practice, we do not want to round $W$ just to the integers, but instead to scale it, shift it, and round it a finite subset corresponding to a $b$-bit integer. To do this, the "real" LDLQ algorithm uses a clamp operation to restrict the range of quantized values. Is LDLQ still optimal when this small change is made? It turns out that the answer is *no*, as the following concrete example illustrates.

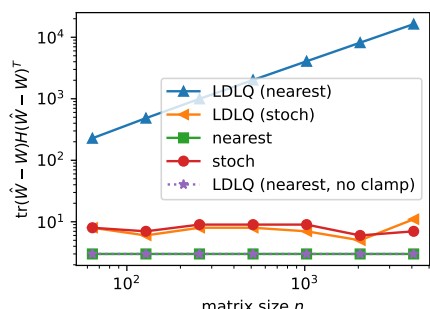

Figure 4: LDLQ underperforms.

**Finite Grid Counterexample.** Figure 4 illustrates the behavior of LDLQ and other rounding methods—when restricted via clamping to a finite 4-bit grid $[0, 15]$—on a particular example where $H$ is a (cleverly chosen) small perturbation of $(I_n + \mathbf{1}_{n \times n} - e_n e_n^T)/n$, and $W$ has $m = 16$ and is a small perturbation of $\mathbf{1}_{m \times n}/2$. Details of the setup appear in Supplement C. The figure shows that clamped LDLQ with nearest rounding is asymptotically worse, and the clamping to the finite grid is what causes it to be worse in this case.

Note that in our experiments in practice, OPTQ has been shown to soundly beat nearest rounding. This clamping issue does not seem to arise in practice; however, since it is *possible* we do need to take it into account to prove useful end-to-end bounds.

**A Procedure With a Bound.** In order to address the above issues in theory, here we describe a method that acts to restrict the value of $|\hat{W}_{ij} - W_{ij}|$, so that the rounded weights will remain inside the grid if $W$ is sufficiently far inside. We do this via the optimization problem with hyperparameter $c$

$$
\begin{aligned}
\text{minimize: } & \operatorname{tr}\left(HR^T R\right) \\
\text{over: } & R \text{ unit upper triangular} \\
\text{subject to: } & e_i^T R^T R e_i \leq 1 + c, \ \forall i \in \{1, \dots, n\}.
\end{aligned}
\tag{7}
$$

Our "fixed" algorithm solves this convex problem (e.g. with ADMM), then runs QuIP using stochastic rounding and $U = R^{-1} - I$ in place of the LDL decomposition. Observe that for sufficiently large $c$, this is exactly equivalent to base QuIP, since the solution of that optimization problem is given by the LDL decomposition when the constraint is dropped. Doing this (the full algorithm is given in the supplemental) yields the following theorem.

**Theorem 7.** *Suppose that we run Algorithm 5 (Supplement) to quantize a matrix $W \in \mathbb{R}^{m \times n}$ by solving the objective (7). Then there exists an assignment of the algorithm's hyperparameters $c$ and $\rho$ such that with probability at least $1 - \delta$, all the quantized weights will be in range (no overflow or need for clipping) and*

$$\operatorname{tr}\left( (\hat{W} - W)H(\hat{W} - W)^T \right) = \tilde{\mathcal{O}}\left( \frac{1}{n^2 4^b} \operatorname{tr}\left( H^{1/2} \right)^2 \|W\|_F^2 \right).$$

In practice, because clamping rarely causes issues, and because of the significant additional compute needed to solve this program, we always just use QuIP as described in the previous sections, which is equivalent to setting $c$ large and using nearest rounding.

## 6   Experiments

**Overview.** We quantize the OPT [35] family of models (up to 66B parameters) and Llama 2 70B [27] using various quantization and processing methods. QuIP is superior to OPTQ and other baselines across all model sizes and evaluation tasks. Most interestingly, incoherence processing yields excellent performance using as little as two bits per weight when paired with any of the quantization methods we consider (including nearest rounding). Two-bit quantization with QuIP is viable at even moderate model sizes (1B parameters), a regime where other two-bit quantization methods fail. At the largest model sizes, the difference between 2-bit and 16-bit weight performance becomes small. We compare the throughput of QuIP with OPTQ's efficient implementation on language generation and show that it is not much slower. Additional results on the effectiveness of the proxy loss, unbiased rounding, and Algorithm 5 are presented in the Supplement C.

**Setup.** The experimental infrastructure is built on top of OPTQ's [8] repository which is implemented in PyTorch [23]. We quantize the HuggingFace implementations of the OPT and Llama 2 model families. All models are quantized on a single GPU, with up to 48GB of memory. Our calibration set is the same as OPTQ; 128 random 2048 token segments from the C4 dataset [25] consisting of generic text data from crawled websites. Therefore, no task-specific data is viewed when quantizing. Following OPTQ, quantization is performed one Transformer block at a time: loaded into GPU memory, the Hessian computed, and then the weights quantized. The current block's inputs are then passed through the quantized block to produce inputs for the following block. The Hessian is computed from the quantized Transformer up to that point rather than from the full precision model; like OPTQ, we find this improves quantization. Further details on the setup can be found in Supplement C, including a description of the computational resources used to perform the experiments.

**Methods.** We evaluate compositions of several quantization and pre/post processing methods. For quantization methods, we evaluate nearest rounding, LDLQ (or OPTQ), and two variations. LDLQ-RG re-orders the weights based on $\operatorname{diag}(H)$ to modify the quantization order and adds further greedy updates to the proxy. "Greedy" performs the greedy updates only. We evaluate the baseline preprocessing from OPTQ which adds $H \leftarrow H + \alpha * \operatorname{mean}(\operatorname{diag}(H))I$ for numerical stability. We also evaluate our incoherence processing in Algorithms 1 and 2, denoted as "IncP". With this notation QuIP = LDLQ + IncP, and QuIP-RG = LDLQ-RG + IncP.

**Datasets.** We evaluate on the following language generation tasks: WikiText2 [17], Penn Treebank (PTB) [16], and C4. We also evaluate on zero-shot tasks, including LAMBADA (LAMB) [21], ARC Easy (ArcE) [1], PiQA [26], and StoryCloze (SC) [18]. See Supplement C for the full set of results.

**Main Results.** QuIP is the first PTQ procedure to achieve good quantization at two bits per weight, across a variety of LLM sizes and evaluation tasks. In Figure 5 we compare QuIP and OPTQ when quantizing to 2 and 3 bits per weight (4-bit quantization works equally well for both methods); we evaluate OPT models (up to 66B) on PTB, C4, ARC Easy, and LAMBADA. QuIP is superior to OPTQ across the model sizes and evaluation tasks. At three bits, QuIP matches the full precision model reasonably well. At two bits and for larger LLMs (>2B parameters), QuIP begins to approach the performance of the full precision model. As model size increases, so does the quality of QuIP's

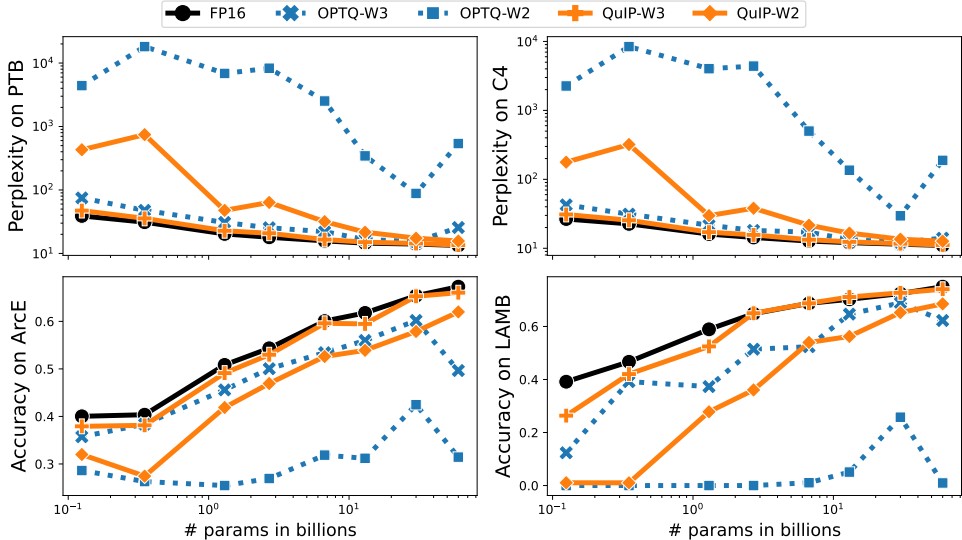

Figure 5: Quantizing OPT models up to 66B parameters. Our method QuIP is the first PTQ procedure to achieve good quantization at 2 bits per weight, across a variety of model sizes and evaluation tasks.

| | OPTQ | | | | | QuIP (Ours) | | | | |
|---|---|---|---|---|---|---|---|---|---|---|
| WBits | Wiki↓ | C4↓ | ArcE↑ | PiQA↑ | SC↑ | Wiki↓ | C4↓ | ArcE↑ | PiQA↑ | SC↑ |
| 16 | 3.319 | 5.709 | 59.72 | 80.90 | 79.95 | 3.319 | 5.709 | 59.72 | 80.90 | 79.95 |
| 4 | 3.596 | 5.905 | 58.96 | 80.52 | 79.12 | 3.531 | 5.869 | 59.81 | 80.47 | 79.63 |
| 3 | 4.907 | 7.099 | 54.38 | 78.56 | 77.72 | 3.853 | 6.135 | 59.81 | 80.25 | 79.31 |
| 2 | 123.908 | 70.541 | 25.34 | 50.54 | 51.75 | **6.326** | **8.937** | **54.38** | **75.08** | **75.37** |

Table 1: Quantizing Llama 2 70B with QuIP and OPTQ, and evaluating on language generation and zeroshot tasks. Our incoherence processing enables a step function change in quantization at 2 bits.

| | Baseline Processing | | | | | Incoherence Processing (Ours) | | | | |
|---|---|---|---|---|---|---|---|---|---|---|
| WBits | Wiki↓ | PTB↓ | C4↓ | ArcE↑ | LAMB↑ | Wiki↓ | PTB↓ | C4↓ | ArcE↑ | LAMB↑ |
| 16 | 9.56 | 14.04 | 11.45 | 65.40 | 72.40 | 9.56 | 14.04 | 11.45 | 65.40 | 72.40 |
| | | | OPTQ | | | | | QuIP | | |
| 4 | 9.59 | 14.22 | 11.56 | 64.77 | 72.39 | 9.60 | 14.18 | 11.50 | 65.32 | 73.20 |
| 3 | 10.32 | 15.36 | 12.23 | 60.19 | 68.89 | 9.79 | 14.37 | 11.66 | 65.28 | 72.68 |
| 2 | 71.70 | 88.19 | 29.59 | 42.47 | 25.77 | **11.48** | 17.40 | 13.55 | 57.87 | **65.24** |
| | | | LDLQ-RG | | | | | QuIP-RG | | |
| 4 | 9.64 | 14.20 | 11.56 | 63.76 | 71.94 | 9.66 | 14.11 | 11.51 | 64.86 | 71.86 |
| 3 | 10.31 | 15.15 | 12.15 | 63.43 | 69.78 | 9.75 | 14.44 | 11.68 | 63.51 | 71.53 |
| 2 | 49.40 | 73.45 | 29.12 | 41.20 | 26.35 | 11.68 | **16.94** | 13.44 | **59.51** | 62.31 |
| | | | Greedy | | | | | Greedy + IncP | | |
| 4 | 9.69 | 14.33 | 11.59 | 63.09 | 72.37 | 9.72 | 14.23 | 11.52 | 65.99 | 71.71 |
| 3 | 13.63 | 23.05 | 16.30 | 50.51 | 56.76 | 9.92 | 14.45 | 11.71 | 63.80 | 71.38 |
| 2 | 4816.6 | 3473.81 | 3183.2 | 26.30 | 0.00 | 11.59 | 17.39 | **13.30** | 58.80 | 64.47 |
| | | | Near | | | | | Near + IncP | | |
| 4 | 10.77 | 15.41 | 13.52 | 61.28 | 70.42 | 9.77 | 14.16 | 11.53 | 64.06 | 71.41 |
| 3 | 1564.9 | 1526.2 | 1808.2 | 34.47 | 1.73 | 9.89 | 14.49 | 11.74 | 64.06 | 71.41 |
| 2 | 41547.8 | 34348.6 | 24815.7 | 25.80 | 0.00 | 12.04 | 18.12 | 14.11 | 56.36 | 60.64 |

Table 2: Quantizing OPT-30b with various quantization and processing methods, and evaluating on language generation and zeroshot tasks. Our incoherence processing enables a step function change in quantization at 2 bits, across all rounding methods.

| Wbits | Rescale | Incoherence | Rescale+Incoherence | Rescale+Incoherence+Quant Range |
|---|---|---|---|---|
| 4 | 24.30 | 24.32 | 24.05 | 23.89 |
| 3 | 32.62 | 42.28 | 31.32 | 26.36 |

Table 3: Ablating sub-steps of QuIP's incoherence processing, see Algorithm 1. Perplexities are averaged over WikiText2, PTB, and C4 for OPT-350m.

2-bit quantization. We provide plots on the remaining datasets in Supplement C. Note that the dip in OPTQ on OPT-66B is documented in their paper.

Table 1 shows the results of quantizing Llama 2 70B using QuIP and OPTQ. Again, QuIP achieves good quantization at two bits while OPTQ does not.

**Incoherence Processing Ablation.** Table 2 shows all combinations of quantization and processing methods evaluated on OPT-30B. At lower weight bits, QuIP's incoherence processing dramatically improves the performance of all quantization methods, across all evaluation tasks. Remarkably, all quantization methods—even nearest—are viable at two bits with our incoherence processing. Our modifications in QuIP-RG sometimes give an improvement over QuIP, but further study is required to evaluate these modifications. Figures for OPT-125M to 13B are in Supplement C.

| Method | Throughput |
|---|---|
| QuIP | 81ms |
| OPTQ | 53ms |

Table 4: Average per-token throughput (batch size 1) when generating sequences of length 128 with OPT-66B on an A6000 GPU.

**Throughput Comparison.** We evaluate the additional overhead of our incoherence processing during model inference by modifying OPTQ's efficient forward pass. OPTQ's implementation contains a quantized-matrix full-precision-vector product kernel and was shown to offer speedups over a FP16 baseline. Our incoherence processing additions are performed in PyTorch. Table 4 shows that our QuIP implementation is about $1.5\times$ slower than OPTQ.

**Further Ablation.** QuIP's incoherence processing contains several sub-steps. Table 3 shows their relative contributions; all are necessary for the full improvement. Table 5 shows that the random permutation step within the fast orthogonal multiplication also significantly reduces perplexity.

## 7 Conclusion

This paper introduced quantization with incoherence processing (QuIP), an algorithm consisting of (1) an optimal adaptive rounding procedure which minimizes a quadratic proxy of the weight error, and (2) efficient pre- and post-processing to ensure the incoherence of the weight and Hessian matrices by multiplying them by a Kronecker product of random orthogonal matrices. We showed that QuIP quantization is optimal in a general class of adaptive rounding methods with linear feedback; this theoretical analysis is the first for any quantization algorithm that scales to LLM-sized models.

| Wbits | $\Delta$Perplexity from random permute$\downarrow$ |
|---|---|
| 4 | -0.22 |
| 3 | -9.96 |
| 2 | -74.2 |

Table 5: Ablating random permutation within fast orthogonal multiplication. Differences in perplexity are averaged over WikiText2, PTB, and C4 for OPT-125m.

Empirically, QuIP achieves the first viable two-bit quantization results for LLMs, especially at large model sizes, hinting at the feasibility of accurate 2-bit inference in LLMs.

## Acknowledgements and Disclosure of Funding

This work was partially funded by the National Science Foundation under awards DGE-1922551, CAREER awards 2046760 and 2145577, by the National Institute of Health under award MIRA R35GM151243, and a gift from CISCO.

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

# A    Checklist

## A.1    Broader Impacts

Our work pushes the quantization of large language models into the 2 bits per weight regime. Our aim is to drive foundational research on theoretical and empirical aspects of quantization. The ultimate goal is to enable more powerful LLMs to run more efficiently. However our work is unaware to what ends those LLMs are used.

## A.2    Limitations

The adaptive rounding [3] proxy objective considers each layer in isolation; it remains to be seen what other computationally tractable proxies could improve quantization. For example quantization methods do exist which consider interactions between layers, but so far have been too computationally expensive to be applied to the largest open LLMS.

## A.3    Experiments, Reproducibility

Our code is included in the Supplement. See the included README for instructions on how to reproduce the various experiments, including random seeds. The code also downloads all datasets used to quantize or evaluate the models.

# B    Additional Method Clarifications

## B.1    Subsection 4.2 (Incoherence-Based Heuristics)

Line 4 diagonally rescales $W$ and $H$ to minimize $\ell(\hat{W}) \approx \mathrm{tr}(H) \|W\|_F^2$, effectively trading off the spectrum of these matrices to find a minimum. Note to minimize $\mathrm{tr}\left(D^{-1}HD^{-1}\right) \|WD\|_F^2 = \left(\sum_{i=1}^n H_{ii}/D_i^2\right)\left(\sum_{i=1}^n D_i^2\|W_i\|^2\right)$ implies that $D_i = \sqrt{H_{ii}/\|W_i\|}$. Motivated by the incoherence of $W$, Line 6 computes the quantization range depending on the spectrum $\|W\|_F$, instead of the typical $\max_{i,j}|W_{ij}|$. The parameter $\rho$ controls the quantization range; we tune it and find that a value of 2.4 works well across all our experiments. We use $\rho = 2.4$ consistently across all experiments. Our full QuIP procedure is described in Algorithm 3, which contains calls to the pre- and post-processing sub-steps in Algorithms 1 and 2.

## B.2    Subsection 4.2 (Greedy Updates)

In this subsection, we describe the "greedy local search" method mentioned in the main body of the paper in more detail. The basic idea is to iterate over coordinates of the weights in the same order as the initial quantization method, modifying each weight in turn—but still restricting it to be a representable quantized value—so as to minimize the proxy loss while keeping the other weights fixed. These greedy updates amount to coordinate descent on the proxy loss, but restricted to the quantization grid. Greedy updates can be performed after any initial quantization method, or as a standalone method. When performed after an initial quantization method, greedy local search is a *descent method* because the individual weight updates cannot increase the loss, but when performed alone, these greedy updates are not a descent method because the initial point ($\hat{W} = W$) is not feasible because it contains unquantized values that are off the representable quantization grid. Concretely, a greedy update of weight $(i, j)$ to the grid $\{0, 1, \ldots, 2^b - 1\}$ does the following, where $\ell$ is the proxy loss:

$$\hat{W}_{ij} \leftarrow \arg\min_{z \in \{0,1,\ldots,2^b-1\}} \ell(\hat{W} - e_i e_j^T \hat{W}_{ij} + e_i e_j^T z).$$

(Note that $\hat{W} - e_i e_j^T \hat{W}_{ij} + e_i e_j^T z$ is the result of setting the $(i, j)$th entry of $\hat{W}$ to $z$.) A full pass of greedy updates constitutes $mn$ of these updates performed in the same order as LDLQ. This algorithm is very simple, since it is just greedy coordinate descent. In the rest of this subsection, we will give a bit more intuition about this method by showing how this greedy algorithm falls within our framework of adaptive rounding with linear feedback.

---

**Algorithm 4** Greedy Updates: A Single Pass

---

**Require:** $b \in \mathbb{N}$, $H \in \mathbb{R}^{n \times n}$ SPD, weights $W \in \mathbb{R}^{m \times n}$, initial guess $\tilde{W}$
1: $\hat{W} \leftarrow \tilde{W}$
2: $U \leftarrow (H \odot M) \operatorname{diag}(H)^{-1}$ $\qquad\qquad\qquad$ ▷ $M$ is the strictly upper triangular mask
3: $V \leftarrow W - (\tilde{W} - W)(H \odot M^T) \operatorname{diag}(H)^{-1}$ $\qquad$ ▷ can skip if $\tilde{W} = W$ by setting $V \leftarrow W$
4: **for** $k \in \{1, \ldots, n\}$ **do** $\hat{W}_k \leftarrow \operatorname{clamp}(\mathcal{Q}_{\text{near}}(V_k + (W - \hat{W})U_k), 0, 2^b - 1)$
5: **return** $\hat{W}$

---

An application of greedy local search as a single-pass stand-alone method falls under our Adaptive Rounding with Linear Feedback framework, with the linear feedback set to $U = (H \odot M) \operatorname{diag}(H)^{-1}$, where $M$ is the strictly upper triangular mask and $\odot$ denotes the Hadamard (entrywise) product, as we will derive below. For ease of explanation consider a single (row) weight vector $w \in \mathbb{R}^{1 \times n}$. When looking only at column $j$, the proxy loss from setting $\hat{w}_j$ to $z$ is

$$\ell(\hat{w} - \hat{w}e_j e_j^T + z e_j^T) = (\hat{w} - w)H(\hat{w} - w)^T + 2(z e_j^T - \hat{w}e_j e_j^T)H(\hat{w} - w)^T$$
$$+ (z e_j^T - \hat{w}e_j e_j^T)H(z e_j^T - \hat{w}e_j e_j^T)^T.$$

This is just a quadratic function in $z$, and so its minimum value on the grid $\{0, 1, \ldots, 2^b - 1\}$ will just be its minimum value on $\mathbb{R}$ rounded to that grid. To find this minimum over $\mathbb{R}$, we differentiate to minimize, yielding

$$0 = 2e_j^T H(\hat{w} - w)^T + 2e_j^T H(z e_j^T - \hat{w}e_j e_j^T)^T,$$

and solving for $z$,

$$z = -\frac{(\hat{w} - \hat{w}e_j e_j^T - w)H e_j}{e_j^T H e_j} = \hat{w}e_j - \frac{(\hat{w} - w)H e_j}{e_j^T H e_j}. \tag{8}$$

Since when we use greedy local search as a stand-alone method, we have not updated $\hat{w}_j$ yet, at this point $\hat{w}e_j = we_j$, and so this means that a single step of greedy updates looks like

$$\hat{w}e_j \leftarrow \mathcal{Q}\left(we_j - (\hat{w} - w)\frac{H e_j}{e_j^T H e_j}\right)$$

for $\mathcal{Q}$ referring to nearest rounding with the necessary clamping. Since $\hat{w} - w$ is zero for all entries following the $j$th one, this is equivalent to

$$\hat{w}e_j \leftarrow \mathcal{Q}(we_j - (\hat{w} - w)U e_j)$$

where $U$ is set as $U = (H \odot M) \operatorname{diag}(H)^{-1}$ as above. This shows how this single-pass version of greedy updates fits into our adaptive rounding with linear feedback framework.

Analyzing greedy local search as a post-processing pass is a bit more difficult, but we will see that it can also be written as something like adaptive rounding with linear feedback. Suppose that we do a pass of greedy updates, but our quantized weights start at an initial value $\hat{w} = \tilde{w}$ already quantized from some previous method (e.g. LDLQ). Returning to (8), since we haven't updated $\hat{w}_j$ yet, we'll have

$$z = \tilde{w}e_j - \frac{(\hat{w} - w)H e_j}{e_j^T H e_j}.$$

Now, all the entries of $\hat{w}$ which come *after* $j$ are still the ones from $\tilde{w}$. This means that we can split this up as

$$z = we_j - \frac{(\hat{w} - w)_{:,1:(j-1)}H_{1:(j-1),j} + (\tilde{w} - w)_{:,(j+1):n}H_{(j+1):n,j}}{e_j^T H e_j}$$

where the first part of this sum comes from the entries which we *may* have already updated during this pass, the second comes from the entries which are still equal to their initial values in $\tilde{w}$, and the case of $w_j$ is handled specially, cancelling it with the $\tilde{w}e_j$ term. We can write this more compactly in matrix form as

$$z = we_j - \frac{(\hat{w} - w)(H \odot M)e_j + (\tilde{w} - w)(H \odot M^T)e_j}{e_j^T H e_j},$$

| Model | Processing | Absolute Fractional Rank | Approximate Fractional Rank | $\operatorname{tr}(D)/\operatorname{tr}(H)$ |
|---|---|---|---|---|
| OPT-125m | Baseline | 0.926 (±0.172) | 0.112 (±0.127) | 0.540 (±0.093) |
| | Incoherent | 0.910 (±0.196) | 0.124 (±0.141) | 0.534 (±0.094) |
| OPT-350m | Baseline | 0.916 (±0.180) | 0.047 (±0.032) | 0.445 (±0.100) |
| | Incoherent | 0.908 (±0.183) | 0.059 (±0.062) | 0.440 (±0.106) |
| OPT-1.3b | Baseline | 0.541 (±0.404) | 0.020 (±0.023) | 0.399 (±0.187) |
| | Incoherent | 0.543 (±0.405) | 0.028 (±0.023) | 0.393 (±0.189) |
| OPT-2.7b | Baseline | 0.426 (±0.413) | 0.019 (±0.015) | 0.384 (±0.206) |
| | Incoherent | 0.427 (±0.415) | 0.018 (±0.025) | 0.375 (±0.205) |

Table 6: We compute $H$ in each layer of a given model, and compute the following summary statistics. $\operatorname{tr}(D)/\operatorname{tr}(H)$ decreases as the mode size increases, though the variance also increases. We compute the fraction of nonzero eigenvalues (i.e. absolute), and the fraction of eigenvalues $> 0.01 \cdot \max(\operatorname{eig}(H))$ (i.e. approximate). The fractional rank is $k/n$ for a rank-$k$ matrix $H$ with dimension $n$. Mean and standard deviations are computed across layers in a model.

where $M$ is the strictly upper triangular mask and $\odot$ is elementwise multiplication. This yields a final quantization step of

$$\hat{w}e_j \leftarrow \mathcal{Q}\left(we_j - (\tilde{w} - w)\frac{(H \odot M^T)e_j}{e_j^T He_j} - (\hat{w} - w)\frac{He_j}{e_j^T He_j}\right).$$

So, more generally, if we define $U$ as above, and set

$$V = W - (\tilde{W} - W)(H \odot M^T)\operatorname{diag}(H)^{-1},$$

we can write a single pass of greedy updates in matrix form as

$$\tilde{W} \leftarrow \mathcal{Q}(V + (W - \hat{W})U),$$

which is very close to our rounding with linear feedback form, albeit with the difference that here $V$ is in place of $W$. This is made explicit in the included Greedy Updates Algorithm.

We can use this algorithm both as a whole quantization method (by setting $\tilde{W} = W$) or as a post-processing step (by setting $\tilde{W}$ to the output of some other initial quantization algorithm, such as LDLQ). When we do use it as a post-processing step, we typically run multiple passes of greedy updates (e.g. 10 passes): this involves passing the output of the greedy updates algorithm back in as the input guess $\tilde{W}$ to another run of the greedy updates algorithm, and repeating this multiple times.

## C  Additional Experimental Descriptions and Results

### C.1  Subsections 3.2 and 3.3 (Empirical Properties of $H$ Across OPT-125m to 2.7b)

**Interpreting the exact proxy loss of LDLQ and nearest rounding by empirically comparing** $\operatorname{tr}(D)$ **vs** $\operatorname{tr}(H)$. Theorem 1 gives the average-case proxy loss for LDLQ in terms of $\operatorname{tr}(D)$, where $D$ is from the LDL decomposition of $H$. Lemma 3 gives the average-case proxy loss for standard nearest rounding in terms of $\operatorname{tr}(H)$. We know that LDLQ is better in practice, but comparing these equations is difficult because we need to reason about $\operatorname{tr}(D)$ vs $\operatorname{tr}(H)$. Our paper resolves this difficulty by deriving bounds on the proxy loss for LDLQ in terms of the spectrum of $H$ (with and without incoherence). However we also perform a quick empirical check: if $\operatorname{tr}(D) \ll \operatorname{tr}(H)$, then our theory explains the empirical superiority of LDLQ over nearest rounding (at least on these models). Table 6 gives the ratio $\operatorname{tr}(D)/\operatorname{tr}(H)$ across all layers for OPTQ models 125m to 2.7b; the mean value is always less than $0.55$, and it falls as the model gets larger.

**$H$ is approximately low-rank.** Subsection 3.3 plotted the normalized eigenvalues of $H$ from 3 randomly chosen layers in OPT-2.7b. Table 6 gives much more evidence that $H$ is consistently approximately low-rank. Across each model, we calculate the absolute and approximate fractional rank

of $H$ across all layers in OPT models 125m to 2.7b (explanations in the caption). The approximate fractional rank decreases as model size increases; for OPT-2.7b the fractional rank is $\approx 0.02(\pm 0.02)$.

## C.2 Subsection 5.1 (Empirical Verification of OPTQ Equivalence)

We share a python script in the supplementary code which empirically verifies that our implementation of LDLQ produces quantized values exactly matching OPTQ's [1] implementation. While we prove the equivalence between LDLQ and OPTQ's respective algorithm statements, empirically comparing ours and Frantar et al. [1]'s code ensures that the respective implementations are sufficiently close to their algorithmic statements. Therefore we can be sure that LDLQ and OPTQ are equivalent in their implementation.

## C.3 Subsection 5.2 (Empirical Verification of LDLQ/OPTQ Finite Grid Counterexample)

The following code constructs a weight matrix $W$ and Hessian matrix $H$ where OPTQ performs worse than nearest when rounding to a finite grid.

```
import torch
def make_counterexample(n, d, c=0.01):
    H = torch.ones(n,n) + torch.eye(n)
    H[n-1,n-1] = 1.0
    H[0,1:(n-1)] += 2 * c
    H[1:(n-1),0] += 2 * c
    H[0,n-1] += c
    H[n-1,0] += c
    H[0,0] += 4 * c + n * (c**2)
    W = 0.499 * torch.ones(d,n) + 0.002 * (torch.arange(n) % 2)
    return W, H
```

The intuition behind this counterexample is as follows: we want to quantize many coordinates in $W$ in such a way that OPTQ excepts there to be a very large error correction to quantize the last entry. However, the finite grid restricts this large error correction. Note that we can achieve this poor OPTQ behavior with c=0, but here nearest rounding also does poorly. We make a small perturbation (c=0.01) to make OPTQ round in the wrong direction, but not nearest.

## C.4 Additional Details on the Experimental Setup and Computational Resources

We run experiments on a university cluster managed by a Slurm workload manager which has GPUs with up to 48GB of memory, though larger GPUs are only required for some methods on larger model sizes. Note we use the LAMBADA OpenAI version. When Greedy updates are used, we perform 10 passes over the weights in the same order as LDLQ and OPTQ, except for 5 passes on OPT-30b and OPT-66b. For the incoherence-based quantization range, we tune the parameter $\rho$ and find that a value of 2.4 works well across all model sizes and quantization methods. We use this value for all our experiments.

## C.5 Section 6 (Main Results on Additional Evaluations)

Figure 6 shows additional results for QuIP and OPTQ on WikiText2, PiQA, and StoryCloze when quantizing to 2 and 3 bits per weight. The insights about our method QuIP remain the same after viewing these additional results: QuIP is the first PTQ procedure to achieve good quantization at two bits per weight, across a variety of LLM sizes and evaluation tasks. We evaluate on OPT models (up to 30B); 4-bit quantization works equally well for both methods. QuIP is superior to OPTQ across model sizes and evaluation tasks here.

On WikiText2 2-bit quantization, note that the trend in perplexity for QuIP mirrors the trend in perplexity for OPTQ. We run OPTQ's [1] implementation, though they did not report 2-bit results at this model size. Because OPTQ is equivalent to QuIP's quantization sub-procedure, it thus makes sense that worse performance in the quantization sub-procedure could result in worse overall performance. OPTQ increases perplexity when going from OPT-1.3b to OPT-2.7b. QuIP's perplexity also increases from OPT-1.3b to OPT-2.7b, and is unusually higher than the adjacent OPT-1.3b and

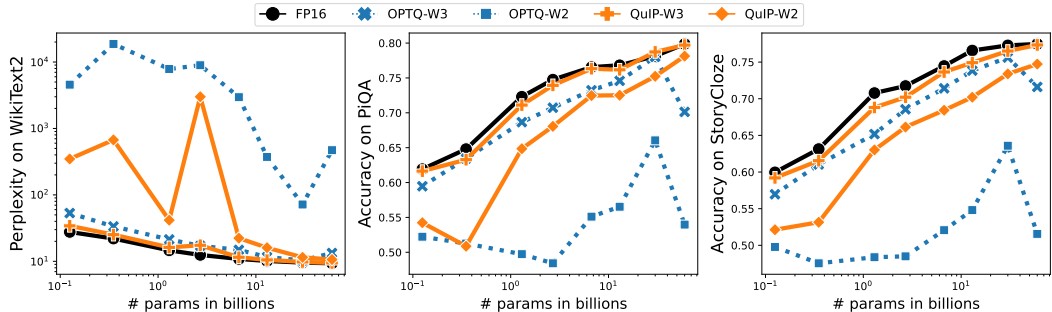

Figure 6: Quantizing OPT models up to 66B parameters. Additional evaluation tasks shown here in the Supplement. Our method QuIP is the first PTQ procedure to achieve good quantization at 2 bits per weight, across a variety of model sizes and evaluation tasks. Note the drop in performance for OPTQ on OPT-66B is documented in their paper.

OPT-6.7b models. However QuIP still beats OPTQ in this setting. Our observations about OPTQ and QuIP on WikiText2 and OPT-2.7b were consistent across multiple independent runs.

### C.6  Section 6 (All Methods, All Model Sizes, All Bit Weights, All Evaluation Tasks)

Tables 7-13 provide results on all combinations of the following: methods, model sizes (OPT 125m-30b), bit weights(4,3,2), and evaluation tasks. Across our extensive array of experiments, we see that incoherence processing always enables a step function change in quantization at 2 bits.

| | **Incoherence Processing — OPT-30b** | | | | | | | | | | | | |
| | Full | QuIP | | | QuIP-RG | | | Greedy+IncP | | | Near+IncP | | |
| | W16 | W4 | W3 | W2 | W4 | W3 | W2 | W4 | W3 | W2 | W4 | W3 | W2 |
| Wiki↓ | 9.56 | 9.60 | 9.79 | 11.48 | 9.66 | 9.75 | 11.68 | 9.72 | 9.92 | 11.59 | 9.77 | 9.89 | 12.04 |
| PTB↓ | 14.04 | 14.18 | 14.37 | 17.40 | 14.11 | 14.44 | 16.94 | 14.23 | 14.45 | 17.39 | 14.16 | 14.49 | 18.12 |
| C4↓ | 11.45 | 11.50 | 11.66 | 13.55 | 11.51 | 11.68 | 13.44 | 11.52 | 11.71 | 13.30 | 11.53 | 11.74 | 14.11 |
| ArcE↑ | 65.40 | 65.32 | 65.28 | 57.87 | 64.86 | 63.51 | 59.51 | 65.99 | 63.80 | 58.80 | 64.06 | 64.06 | 56.36 |
| LAMB↑ | 72.40 | 73.20 | 72.68 | 65.24 | 71.86 | 71.53 | 62.31 | 71.71 | 71.38 | 64.47 | 71.41 | 71.41 | 60.64 |
| PiQA↑ | 78.13 | 78.45 | 78.73 | 75.24 | 78.51 | 78.73 | 76.17 | 77.86 | 77.58 | 75.95 | 78.24 | 77.53 | 75.46 |
| SC↑ | 77.28 | 76.96 | 76.51 | 73.39 | 77.02 | 77.08 | 73.01 | 76.70 | 76.64 | 73.33 | 76.77 | 75.94 | 71.93 |
| | **Baseline Processing — OPT-30b** | | | | | | | | | | | | |
| | Full | OPTQ | | | LDLQ-RG | | | Greedy | | | Near | | |
| | W16 | W4 | W3 | W2 | W4 | W3 | W2 | W4 | W3 | W2 | W4 | W3 | W2 |
| Wiki↓ | 9.56 | 9.59 | 10.32 | 71.70 | 9.64 | 10.31 | 49.40 | 9.69 | 13.63 | 4,817 | 10.77 | 1,565 | 41,548 |
| PTB↓ | 14.04 | 14.22 | 15.36 | 88.19 | 14.20 | 15.15 | 73.45 | 14.33 | 23.05 | 3,474 | 15.41 | 1,526 | 34,349 |
| C4↓ | 11.45 | 11.56 | 12.23 | 29.59 | 11.56 | 12.15 | 29.12 | 11.59 | 16.30 | 3,183 | 13.52 | 1,808 | 24,816 |
| ArcE↑ | 65.40 | 64.77 | 60.19 | 42.47 | 63.76 | 63.43 | 41.20 | 63.09 | 50.51 | 26.30 | 61.28 | 34.47 | 25.80 |
| LAMB↑ | 72.40 | 72.39 | 68.89 | 25.77 | 71.94 | 69.78 | 26.35 | 72.37 | 56.76 | 00.00 | 70.42 | 01.73 | 00.00 |
| PiQA↑ | 78.13 | 78.56 | 78.02 | 66.05 | 78.56 | 77.80 | 64.58 | 78.35 | 70.46 | 49.89 | 77.02 | 56.37 | 49.56 |
| SC↑ | 77.28 | 77.53 | 75.62 | 63.59 | 76.89 | 75.56 | 63.53 | 76.45 | 68.43 | 48.31 | 75.24 | 49.59 | 48.57 |

Table 7: Quantizing **OPT-30b** with all combinations of quantization and pre-post processing methods, evaluating on language generation and zeroshot tasks. Our incoherence processing enables a step function change in quantization at 2 bits, across all rounding methods.

| | Incoherence Processing — OPT-13b | | | | | | | | | | | | |
|---|---|---|---|---|---|---|---|---|---|---|---|---|---|
| | Full | QuIP | | | QuIP-RG | | | Greedy+IncP | | | Near+IncP | | |
| | W16 | W4 | W3 | W2 | W4 | W3 | W2 | W4 | W3 | W2 | W4 | W3 | W2 |
| Wiki↓ | 10.13 | 10.21 | 10.5 | 16.02 | 10.35 | 10.69 | 13.81 | 10.25 | 10.61 | 13.91 | 10.34 | 10.59 | 16.12 |
| PTB↓ | 14.52 | 14.69 | 15.05 | 21.64 | 14.73 | 15.20 | 22.23 | 14.85 | 15.11 | 20.20 | 14.93 | 15.27 | 23.18 |
| C4↓ | 12.06 | 12.16 | 12.39 | 16.60 | 12.18 | 12.43 | 15.62 | 12.21 | 12.42 | 15.19 | 12.26 | 12.56 | 17.37 |
| ArcE↑ | 61.78 | 61.41 | 59.47 | 53.91 | 60.35 | 61.78 | 52.86 | 60.10 | 59.43 | 53.79 | 60.56 | 59.30 | 50.00 |
| LAMB↑ | 70.25 | 72.09 | 71.10 | 56.24 | 69.47 | 69.07 | 55.70 | 70.83 | 68.43 | 56.98 | 68.37 | 67.86 | 46.48 |
| PiQA↑ | 76.82 | 76.61 | 76.17 | 72.52 | 76.55 | 76.22 | 72.74 | 76.33 | 76.17 | 71.87 | 75.08 | 76.66 | 70.73 |
| SC↑ | 76.58 | 75.62 | 74.92 | 70.21 | 75.88 | 75.75 | 70.53 | 75.43 | 75.62 | 72.50 | 74.47 | 75.43 | 68.43 |
| | Baseline Processing — OPT-13b | | | | | | | | | | | | |
| | Full | OPTQ | | | LDLQ-RG | | | Greedy | | | Near | | |
| | W16 | W4 | W3 | W2 | W4 | W3 | W2 | W4 | W3 | W2 | W4 | W3 | W2 |
| Wiki↓ | 10.13 | 10.31 | 11.60 | 372.68 | 10.28 | 11.54 | 213.75 | 10.73 | 13.67 | 8,370 | 11.33 | 3,333 | 186,069 |
| PTB↓ | 14.52 | 14.91 | 16.59 | 344.44 | 14.85 | 16.43 | 220.38 | 15.25 | 18.62 | 7,053 | 16.40 | 2,708 | 121,291 |
| C4↓ | 12.06 | 12.26 | 13.34 | 135.48 | 12.24 | 13.17 | 67.48 | 12.55 | 14.30 | 4,316 | 13.32 | 2,711 | 93,834 |
| ArcE↑ | 61.78 | 64.77 | 60.19 | 42.47 | 60.77 | 58.54 | 32.07 | 56.61 | 51.22 | 25.38 | 61.32 | 31.10 | 25.42 |
| LAMB↑ | 70.25 | 72.39 | 68.89 | 25.77 | 68.72 | 65.30 | 6.58 | 68.12 | 59.36 | 00.02 | 67.22 | 00.06 | 00.00 |
| PiQA↑ | 76.82 | 78.56 | 78.02 | 66.05 | 76.28 | 75.08 | 59.09 | 76.50 | 73.45 | 50.98 | 76.06 | 53.10 | 49.62 |
| SC↑ | 76.58 | 77.53 | 75.62 | 63.59 | 76.32 | 73.52 | 56.33 | 75.68 | 72.44 | 49.40 | 74.41 | 49.71 | 48.70 |

Table 8: Quantizing **OPT-13b** with all combinations of quantization and pre-post processing methods, evaluating on language generation and zeroshot tasks. Our incoherence processing enables a step function change in quantization at 2 bits, across all rounding methods.

| | Incoherence Processing — OPT-6.7b | | | | | | | | | | | | |
|---|---|---|---|---|---|---|---|---|---|---|---|---|---|
| | Full | QuIP | | | QuIP-RG | | | Greedy+IncP | | | Near+IncP | | |
| | W16 | W4 | W3 | W2 | W4 | W3 | W2 | W4 | W3 | W2 | W4 | W3 | W2 |
| Wiki↓ | 10.86 | 10.98 | 11.51 | 22.33 | 11.20 | 11.61 | 23.75 | 11.13 | 11.62 | 19.06 | 11.18 | 11.73 | 18.57 |
| PTB↓ | 15.77 | 15.93 | 16.52 | 31.73 | 15.99 | 16.43 | 45.53 | 15.88 | 16.50 | 35.94 | 16.06 | 16.47 | 27.04 |
| C4↓ | 12.71 | 12.86 | 13.30 | 21.62 | 12.88 | 13.39 | 24.98 | 12.89 | 13.27 | 19.62 | 12.96 | 13.37 | 19.15 |
| ArcE↑ | 60.06 | 59.89 | 59.60 | 52.61 | 59.30 | 59.26 | 53.32 | 59.18 | 58.25 | 51.43 | 59.85 | 57.62 | 50.59 |
| LAMB↑ | 68.72 | 70.00 | 68.74 | 53.97 | 67.38 | 65.77 | 49.91 | 67.65 | 67.18 | 54.80 | 67.26 | 65.86 | 49.49 |
| PiQA↑ | 76.55 | 76.77 | 76.33 | 72.47 | 76.71 | 76.33 | 72.91 | 76.39 | 75.46 | 72.20 | 76.55 | 76.71 | 71.22 |
| SC↑ | 74.47 | 75.18 | 73.65 | 68.43 | 75.05 | 73.33 | 69.51 | 74.35 | 73.77 | 68.94 | 74.22 | 74.09 | 68.75 |
| | Baseline Processing — OPT-6.7b | | | | | | | | | | | | |
| | Full | OPTQ | | | LDLQ-RG | | | Greedy | | | Near | | |
| | W16 | W4 | W3 | W2 | W4 | W3 | W2 | W4 | W3 | W2 | W4 | W3 | W2 |
| Wiki↓ | 10.86 | 11.49 | 14.87 | 2,958 | 11.23 | 12.56 | 739.9 | 11.75 | 39.09 | 16,298 | 12.15 | 6,011 | 20,780 |
| PTB↓ | 15.77 | 16.54 | 22.05 | 2,521 | 16.28 | 18.58 | 1,109 | 16.93 | 66.57 | 10,708 | 18.92 | 5,440 | 14,217 |
| C4↓ | 12.71 | 13.16 | 17.13 | 500.7 | 12.98 | 14.34 | 154.0 | 13.27 | 37.13 | 9,968 | 14.40 | 5,225 | 12,419 |
| ArcE↑ | 60.06 | 58.84 | 53.41 | 31.86 | 59.18 | 55.26 | 33.00 | 54.63 | 32.49 | 26.09 | 58.75 | 25.42 | 25.80 |
| LAMB↑ | 68.72 | 66.18 | 52.36 | 01.07 | 67.46 | 61.89 | 01.79 | 66.19 | 02.56 | 00.00 | 64.53 | 00.00 | 00.00 |
| PiQA↑ | 76.55 | 76.01 | 73.23 | 55.11 | 76.77 | 74.48 | 54.46 | 74.48 | 53.59 | 51.90 | 76.28 | 50.71 | 49.78 |
| SC↑ | 74.47 | 73.71 | 71.42 | 52.07 | 74.09 | 72.37 | 52.45 | 72.82 | 50.99 | 49.40 | 73.58 | 47.87 | 47.80 |

Table 9: Quantizing **OPT-6.7b** with all combinations of quantization and pre-post processing methods, evaluating on language generation and zeroshot tasks. Our incoherence processing enables a step function change in quantization at 2 bits, across all rounding methods.

| | **Incoherence Processing — OPT-2.7b** | | | | | | | | | | | |
|---|---|---|---|---|---|---|---|---|---|---|---|---|
| | Full | QuIP | | | QuIP-RG | | | Greedy+IncP | | | Near+IncP | | |
| | W16 | W4 | W3 | W2 | W4 | W3 | W2 | W4 | W3 | W2 | W4 | W3 | W2 |
| Wiki↓ | 12.47 | 12.39 | 17.44 | 2,998 | 12.58 | 15.07 | 1,676 | 12.68 | 12.96 | 155.6 | 12.79 | 13.79 | 28.98 |
| PTB↓ | 17.97 | 18.42 | 20.79 | 63.59 | 18.43 | 20.49 | 42.05 | 18.34 | 20.03 | 46.28 | 18.43 | 19.51 | 39.23 |
| C4↓ | 14.34 | 14.55 | 15.63 | 38.07 | 14.65 | 15.97 | 27.89 | 14.64 | 15.22 | 26.84 | 14.67 | 15.52 | 27.34 |
| ArcE↑ | 54.34 | 53.28 | 52.99 | 46.93 | 52.02 | 52.36 | 46.93 | 52.90 | 51.73 | 43.14 | 52.61 | 50.93 | 44.11 |
| LAMB↑ | 64.82 | 66.04 | 64.99 | 36.06 | 64.64 | 63.46 | 43.39 | 64.68 | 62.95 | 45.53 | 65.40 | 61.05 | 35.65 |
| PiQA↑ | 74.76 | 74.54 | 73.94 | 68.06 | 73.88 | 73.45 | 68.28 | 74.54 | 73.83 | 68.28 | 73.61 | 73.56 | 67.85 |
| SC↑ | 71.74 | 71.80 | 70.21 | 66.14 | 71.55 | 70.15 | 64.67 | 70.85 | 71.10 | 65.82 | 71.16 | 70.02 | 63.27 |

| | **Baseline Processing — OPT-2.7b** | | | | | | | | | | | |
|---|---|---|---|---|---|---|---|---|---|---|---|---|
| | Full | OPTQ | | | LDLQ-RG | | | Greedy | | | Near | | |
| | W16 | W4 | W3 | W2 | W4 | W3 | W2 | W4 | W3 | W2 | W4 | W3 | W2 |
| Wiki↓ | 12.47 | 12.93 | 17.09 | 8,949 | 12.77 | 16.47 | 7,718 | 12.95 | 18.92 | 9,665 | 16.69 | 15,685 | 10,641 |
| PTB↓ | 17.97 | 19.10 | 25.36 | 8,281 | 19.05 | 23.94 | 7,389 | 19.06 | 28.75 | 8,254 | 32.22 | 14,532 | 10,516 |
| C4↓ | 14.34 | 14.99 | 18.14 | 4,388 | 14.85 | 17.37 | 2,113 | 15.01 | 20.87 | 5,139 | 18.75 | 11,257 | 9,356 |
| ArcE↑ | 54.34 | 52.57 | 50.04 | 26.94 | 52.02 | 48.95 | 25.76 | 52.02 | 43.39 | 25.46 | 52.74 | 26.56 | 27.19 |
| LAMB↑ | 64.82 | 62.00 | 51.43 | 00.00 | 64.04 | 53.25 | 00.00 | 63.50 | 40.75 | 00.00 | 59.15 | 00.00 | 00.00 |
| PiQA↑ | 74.76 | 73.88 | 70.73 | 48.42 | 74.54 | 69.91 | 49.95 | 73.61 | 66.05 | 50.65 | 73.83 | 51.41 | 50.22 |
| SC↑ | 71.74 | 70.91 | 68.56 | 48.50 | 71.42 | 67.79 | 47.17 | 70.66 | 60.53 | 48.44 | 70.59 | 47.42 | 47.55 |

Table 10: Quantizing **OPT-2.7b** with all combinations of quantization and pre-post processing methods, evaluating on language generation and zeroshot tasks. Our incoherence processing enables a step function change in quantization at 2 bits, across all rounding methods.

| | **Incoherence Processing — OPT-1.3b** | | | | | | | | | | | |
|---|---|---|---|---|---|---|---|---|---|---|---|---|
| | Full | QuIP | | | QuIP-RG | | | Greedy+IncP | | | Near+IncP | | |
| | W16 | W4 | W3 | W2 | W4 | W3 | W2 | W4 | W3 | W2 | W4 | W3 | W2 |
| Wiki↓ | 14.62 | 14.88 | 16.21 | 41.64 | 16.49 | 17.76 | 42.37 | 16.75 | 17.11 | 48.69 | 16.43 | 17.83 | 56.56 |
| PTB↓ | 20.29 | 20.87 | 22.76 | 47.72 | 21.93 | 23.25 | 50.17 | 22.11 | 23.76 | 54.46 | 22.19 | 24.82 | 80.40 |
| C4↓ | 16.07 | 16.38 | 17.12 | 29.78 | 17.53 | 18.44 | 31.49 | 17.60 | 18.54 | 34.10 | 17.74 | 19.03 | 45.56 |
| ArcE↑ | 50.84 | 50.72 | 49.12 | 41.88 | 49.54 | 48.82 | 41.20 | 49.66 | 48.74 | 41.08 | 48.61 | 46.59 | 38.64 |
| LAMB↑ | 58.92 | 56.36 | 52.47 | 27.81 | 51.62 | 48.36 | 27.27 | 49.95 | 48.38 | 19.21 | 49.76 | 51.12 | 20.20 |
| PiQA↑ | 72.31 | 71.22 | 71.11 | 64.85 | 71.06 | 70.24 | 63.33 | 71.00 | 70.35 | 63.66 | 71.16 | 69.80 | 62.51 |
| SC↑ | 70.78 | 70.08 | 68.81 | 63.02 | 69.00 | 68.05 | 63.14 | 68.49 | 67.92 | 62.64 | 69.13 | 67.79 | 58.43 |

| | **Baseline Processing — OPT-1.3b** | | | | | | | | | | | |
|---|---|---|---|---|---|---|---|---|---|---|---|---|
| | Full | OPTQ | | | LDLQ-RG | | | Greedy | | | Near | | |
| | W16 | W4 | W3 | W2 | W4 | W3 | W2 | W4 | W3 | W2 | W4 | W3 | W2 |
| Wiki↓ | 14.62 | 15.59 | 21.35 | 7,856 | 15.36 | 20.22 | 7,739 | 15.58 | 22.68 | 9,786 | 47.62 | 12,658 | 11,690 |
| PTB↓ | 20.29 | 22.03 | 30.74 | 6,858 | 21.85 | 30.10 | 5,368 | 22.00 | 35.18 | 8,441 | 73.51 | 14,705 | 11,690 |
| C4↓ | 16.07 | 16.96 | 21.59 | 4,028 | 16.70 | 20.21 | 2,123 | 16.96 | 22.11 | 5,129 | 27.20 | 6,415 | 8,360 |
| ArcE↑ | 50.84 | 49.33 | 45.58 | 25.46 | 48.95 | 45.41 | 26.68 | 48.19 | 42.42 | 26.01 | 42.80 | 27.82 | 25.13 |
| LAMB↑ | 58.92 | 57.03 | 37.32 | 00.00 | 58.45 | 41.08 | 00.02 | 59.15 | 40.97 | 00.00 | 36.91 | 00.00 | 00.00 |
| PiQA↑ | 72.31 | 70.73 | 68.66 | 49.73 | 70.40 | 67.95 | 52.18 | 70.67 | 66.43 | 50.87 | 67.74 | 51.41 | 49.78 |
| SC↑ | 70.78 | 70.15 | 65.18 | 48.38 | 70.34 | 66.45 | 49.27 | 70.40 | 64.48 | 48.76 | 59.13 | 47.87 | 48.25 |

Table 11: Quantizing **OPT-1.3b** with all combinations of quantization and pre-post processing methods, evaluating on language generation and zeroshot tasks. Our incoherence processing enables a step function change in quantization at 2 bits, across all rounding methods.

|  | **Incoherence Processing — OPT-350m** | | | | | | | | | | | |
| | Full | QuIP | | | QuIP-RG | | | Greedy+IncP | | | Near+IncP | | |
| | W16 | W4 | W3 | W2 | W4 | W3 | W2 | W4 | W3 | W2 | W4 | W3 | W2 |
|---|---|---|---|---|---|---|---|---|---|---|---|---|---|
| Wiki↓ | 22.00 | 22.5 | 25.19 | 672.3 | 23.57 | 25.54 | 418.0 | 23.14 | 25.38 | 239.9 | 23.41 | 27.86 | 1,444 |
| PTB↓ | 31.07 | 32.57 | 35.65 | 744.2 | 32.46 | 37.00 | 587.4 | 33.10 | 37.07 | 301.0 | 33.32 | 39.49 | 1,354 |
| C4↓ | 22.59 | 23.23 | 25.48 | 320.0 | 23.45 | 25.50 | 215.4 | 23.43 | 25.48 | 124.1 | 23.81 | 27.41 | 880.2 |
| ArcE↑ | 40.36 | 39.44 | 38.13 | 27.44 | 39.31 | 38.47 | 29.67 | 39.77 | 40.24 | 30.64 | 38.89 | 38.76 | 28.41 |
| LAMB↑ | 46.67 | 46.89 | 42.03 | 01.03 | 43.04 | 39.80 | 04.99 | 42.44 | 40.62 | 06.38 | 41.47 | 34.45 | 00.08 |
| PiQA↑ | 64.80 | 64.47 | 63.28 | 50.87 | 64.25 | 63.17 | 54.79 | 64.42 | 64.25 | 55.01 | 64.15 | 63.00 | 52.23 |
| SC↑ | 63.14 | 62.13 | 61.55 | 53.15 | 61.74 | 61.23 | 51.43 | 62.83 | 61.62 | 53.28 | 62.38 | 61.49 | 50.22 |
|  | **Baseline Processing — OPT-350m** | | | | | | | | | | | |
| | Full | OPTQ | | | LDLQ-RG | | | Greedy | | | Near | | |
| | W16 | W4 | W3 | W2 | W4 | W3 | W2 | W4 | W3 | W2 | W4 | W3 | W2 |
| Wiki↓ | 22.00 | 24.16 | 33.51 | 18,687 | 23.77 | 31.87 | 10,446 | 27.01 | 137.3 | 23,952 | 25.94 | 64.56 | 23,668 |
| PTB↓ | 31.07 | 34.17 | 47.69 | 18,161 | 33.35 | 44.38 | 8,508 | 40.39 | 153.5 | 15,176 | 36.78 | 87.22 | 28,881 |
| C4↓ | 22.59 | 24.71 | 31.26 | 8,418 | 24.10 | 29.86 | 3,064 | 27.84 | 73.59 | 9,099 | 26.21 | 55.15 | 17,094 |
| ArcE↑ | 40.36 | 38.43 | 38.38 | 26.30 | 39.06 | 37.42 | 25.46 | 38.34 | 31.06 | 24.33 | 38.68 | 36.11 | 25.88 |
| LAMB↑ | 46.67 | 45.60 | 39.20 | 00.00 | 45.26 | 32.54 | 00.02 | 51.45 | 16.63 | 00.00 | 40.66 | 27.46 | 00.00 |
| PiQA↑ | 64.80 | 64.04 | 63.44 | 51.25 | 65.13 | 61.97 | 49.67 | 63.49 | 55.44 | 50.60 | 63.38 | 60.55 | 51.58 |
| SC↑ | 63.14 | 63.78 | 61.04 | 47.55 | 62.57 | 60.53 | 48.95 | 61.36 | 54.87 | 48.44 | 63.02 | 56.84 | 48.95 |

Table 12: Quantizing **OPT-350m** with all combinations of quantization and pre-post processing methods, evaluating on language generation and zeroshot tasks. Our incoherence processing enables a step function change in quantization at 2 bits, across all rounding methods.

|  | **Incoherence Processing — OPT-125m** | | | | | | | | | | | |
| | Full | QuIP | | | QuIP-RG | | | Greedy+IncP | | | Near+IncP | | |
| | W16 | W4 | W3 | W2 | W4 | W3 | W2 | W4 | W3 | W2 | W4 | W3 | W2 |
|---|---|---|---|---|---|---|---|---|---|---|---|---|---|
| Wiki↓ | 27.66 | 33.35 | 34.22 | 347.4 | 31.51 | 42.94 | 361.8 | 30.65 | 55.54 | 230.8 | 31.93 | 37.57 | 397.5 |
| PTB↓ | 38.99 | 40.80 | 47.34 | 430.3 | 43.28 | 51.69 | 414.1 | 41.96 | 48.79 | 250.6 | 43.08 | 52.20 | 441.9 |
| C4↓ | 26.56 | 27.63 | 30.92 | 177.4 | 28.74 | 33.54 | 159.0 | 28.82 | 31.41 | 99.01 | 29.28 | 33.88 | 224.0 |
| ArcE↑ | 40.03 | 38.89 | 37.92 | 31.99 | 39.27 | 38.26 | 31.36 | 38.80 | 37.67 | 33.21 | 38.55 | 37.42 | 32.91 |
| LAMB↑ | 39.16 | 33.03 | 26.37 | 01.05 | 33.75 | 16.96 | 02.17 | 37.78 | 25.34 | 04.66 | 35.65 | 25.21 | 01.82 |
| PiQA↑ | 61.92 | 61.64 | 61.64 | 54.24 | 61.64 | 61.92 | 55.44 | 61.10 | 60.83 | 56.47 | 61.43 | 61.10 | 53.48 |
| SC↑ | 59.96 | 60.03 | 59.20 | 52.13 | 59.07 | 59.26 | 51.94 | 60.15 | 59.52 | 54.04 | 59.13 | 58.88 | 53.41 |
|  | **Baseline Processing — OPT-125m** | | | | | | | | | | | |
| | Full | OPTQ | | | LDLQ-RG | | | Greedy | | | Near | | |
| | W16 | W4 | W3 | W2 | W4 | W3 | W2 | W4 | W3 | W2 | W4 | W3 | W2 |
| Wiki↓ | 27.66 | 31.44 | 53.26 | 4,563 | 32.29 | 53.25 | 3,704 | 77.80 | 1,791 | 3,707 | 37.14 | 1,293 | 5,375 |
| PTB↓ | 38.99 | 45.31 | 74.79 | 4,410 | 45.56 | 75.85 | 3,596 | 101.1 | 1,403 | 4,622 | 53.93 | 1,418 | 4,267 |
| C4↓ | 26.56 | 29.13 | 42.55 | 2,260 | 29.40 | 41.77 | 1,820 | 65.54 | 809.5 | 1,897 | 33.90 | 836.5 | 3,665 |
| ArcE↑ | 40.03 | 38.51 | 35.73 | 28.62 | 39.02 | 36.36 | 27.19 | 34.05 | 26.43 | 27.15 | 36.66 | 30.39 | 26.01 |
| LAMB↑ | 39.16 | 33.69 | 12.36 | 00.00 | 33.26 | 15.00 | 00.00 | 12.25 | 00.00 | 00.00 | 18.22 | 00.08 | 00.00 |
| PiQA↑ | 61.92 | 60.83 | 59.47 | 52.23 | 61.70 | 59.58 | 50.05 | 57.62 | 49.29 | 50.49 | 61.43 | 55.88 | 51.20 |
| SC↑ | 59.96 | 58.88 | 56.97 | 49.78 | 59.20 | 57.03 | 48.95 | 50.99 | 47.55 | 48.82 | 59.96 | 50.03 | 47.93 |

Table 13: Quantizing **OPT-125m** with all combinations of quantization and pre-post processing methods, evaluating on language generation and zeroshot tasks. Our incoherence processing enables a step function change in quantization at 2 bits, across all rounding methods.

## C.7 Section 6 (Evaluating the Effectiveness of the Proxy Objective)

In Table 14 we show the proxy loss of the four quantization methods we evaluate, evaluated over OPT models 125m to 2.7b. The proxy is averaged over models proxy losses normalized by their model dimension; we use $H$ matrices computed as a result of OPTQ and nearest rounding. We do not conduct any processing in the proxy evaluation; this is an evaluation of the rounding methods only. Trends in the proxy reflect end-to-end results. OPTQ/LDLQ, LDLQ-RG, and Greedy are roughly equivalent at 2 bits, and do better than Nearest.

## C.8 Section 6 (Evaluating Unbiased Rounding in LDLQ/OPTQ)

Note in our formulation for Adaptive Rounding with Linear feedback, the $\mathcal{Q}$ subroutine could be biased, or unbiased. It is typical to perform biased rounding in practice; here we investigate if there is

| WBits | LDLQ/OPTQ | LDLQ-RG | Greedy | Near |
|---|---|---|---|---|
| 4 | 104.09 | 105.23 | 120.74 | 301.18 |
| 3 | 529.53 | 475.25 | 537.98 | 1,308.05 |
| 2 | 2,554.89 | 2,291.02 | 2,587.17 | 5,971.69 |

Table 14: Weighted average of proxy Loss $\operatorname{tr}\left((\hat{W}-W)H(\hat{W}-W)^T\right)$ over OPT models 125m to 2.7b. Proxy is averaged over models normalized by their model dimension (768, 1024, 2048, 2560) respectively, to ensure proxy loss is comparable across models of different size. We do not conduct any processing in the proxy evaluation. Trends in the proxy largely reflect end-to-end results: at 2 bits OPTQ, LDLQ-RG, and Greedy are roughly equivalent, and all do better than nearest.

| | AVERAGE(Perplexity Unbiased - Perplexity Biased) on Wiki, PTB, C4 ($\downarrow$) | | | | | | | |
|---|---|---|---|---|---|---|---|---|
| | Incoherence Processing | | | | Baseline Processing | | | |
| WBits | 125m | 350m | 1.3b | 2.7b | 125m | 350m | 1.3b | 2.7b |
| 4 | 1.23 | 0.73 | 0.79 | 0.19 | 27.81 | 5.58 | 1.62 | 0.87 |
| 3 | 13.26 | 7.79 | 2.14 | 4.66 | 880.4 | 499.4 | 28.63 | 16.23 |
| 2 | 2,501 | 18,732 | 544.8 | 2,251 | 241.3 | 17,945 | 4,831 | 3,798 |

Table 15: Average perplexity difference (i.e. unbiased - biased) for LDLQ/OPTQ on WikiText2, PTB, and C4. That is, we can run LDLQ with the $\mathcal{Q}$ subroutine as stochastic rounding, instead of nearest. The average difference is positive, meaning that unbiased rounding performs worse than biased (i.e. nearest) across OPT models 125m to 2.7b. Note the magnitude of the gap increases at lower bits.

any benefit to switching to unbiased rounding schemes. Table 15 computes the average perplexity difference (i.e. $\mathrm{unbiased} - \mathrm{biased}$) for LDLQ/OPTQ on WikiText2, PTB, and C4. That is, we run LDLQ with the $\mathcal{Q}$ subroutine as stochastic rounding, instead of nearest. The average difference is positive (and large for 2 and 3 bits), meaning that unbiased rounding performs worse than biased (i.e. nearest) across OPT models 125m to 2.7b. These results indicate that in practice, we want to stick with biased rounding schemes.

## C.9 Section 6 (Evaluating Algorithm 5 Which Accounts for Clamping)

| Model | WBits | Incoherence Processing (ours) | | | Baseline Processing | | |
|---|---|---|---|---|---|---|---|
| | | Wiki | PTB | C4 | Wiki | PTB | C4 |
| OPT-1.3b | 4 | 16.54 | 22.12 | 17.58 | 15.43 | 21.92 | 16.80 |
| | 3 | 18.27 | 23.96 | 18.66 | 20.45 | 28.86 | 20.68 |
| | 2 | 38.13 | 51.78 | 31.09 | 6,438.75 | 6,099.27 | 2,057.71 |
| OPT-350m | 4 | 23.19 | 32.55 | 23.48 | 23.71 | 33.73 | 24.29 |
| | 3 | 25.54 | 36.74 | 25.52 | 33.01 | 45.15 | 30.09 |
| | 2 | 286.71 | 367.26 | 144.08 | 8,006.22 | 7,445.70 | 2,317.18 |
| OPT-125m | 4 | 32.04 | 44.56 | 29.08 | 32.59 | 41.95 | 28.67 |
| | 3 | 40.66 | 51.90 | 32.91 | 50.73 | 74.14 | 41.04 |
| | 2 | 1,649.83 | 240.86 | 136.55 | 3,714.11 | 4,703.76 | 1,848.72 |

Table 16: Quantizing OPT models using Algorithm 5 evaluated on WikiText2, PTB, and C4. At 2 bits and incoherence processing, we see improvements over LDLQ and LDLQ-RG on OPT-125m and OPT-350m, but diminishing improvements on OPT-1.3b. Due to Algorithm 5's relatively equivalent performance relative to QuIP at OPT-1.3b, and due to this algorithm's increased computational cost, we decide not to user it.

Table 16 shows results from using Algorithm 5 to quantize OPT models 125m to 1.3b, with incoherence processing and baseline processing. At 2 bits and incoherence processing, we observe modest improvements over QuIP in terms of perplexity on OPT models 125m and 350m. However, at the

larger OPT-1.3b QuIP beats Algorithm 5 on 2/3 language generation tasks. In addition, Algorithm 5 is computationally more work to run. Therefore we decide not to use it.

Another observation: in practice, we don't seem to encounter constructions of $W$ and $H$ that are bad for LDLQ/OPTQ. Therefore this "clamping" issue seems to not be an issue in practice, especially as model size increases.

# D    Proofs for Section 3 (Quantization With Incoherence Processing: Adaptive Rounding Step )

### Subsection 3.2 (Deriving the Optimality of the LDLQ Adaptive Rounding Procedure)

**Theorem 1.** LDLQ *is worst and average-case optimal amongst rounding methods which specify the linear feedback $U$ as a function of $H$ (not of $W$), and when rounding to the integers. That is, for all rounding methods $\mathcal{A}$ in the class described by Eq. (2), for all positive semi-definite $H$, and for $\mathcal{Q}$ as either nearest or stochastic rounding,*

$$\frac{m}{4}\operatorname{tr}(D) = \mathcal{L}_{\text{worst}}(\text{LDLQ}, H) \leq \mathcal{L}_{\text{worst}}(\mathcal{A}, H) \ \text{ and } \ \frac{m}{c}\operatorname{tr}(D) = \mathcal{L}_{\text{avg}}(\text{LDLQ}, H) \leq \mathcal{L}_{\text{avg}}(\mathcal{A}, H),$$

*where $D$ is the matrix from the LDL decomposition of $H$, and $c = 12$ for nearest, $c = 6$ for stochastic.*

*Proof.* Let $X$ be the strictly upper triangular matrix associated with the rounding procedure $\mathcal{A}$ such that $U \leftarrow X$ in Eq. (2). Let $B \equiv (X + I)^{-1}(\grave{U} + I)$ where $\grave{U}$ is from the LDL decomposition of $H$ in Eq. (4). The proxy loss is then,

$$\operatorname{tr}\left((\mathcal{A}(W, H) - W)H(\mathcal{A}(W, H))^T\right) \overset{(3),(4)}{=} \operatorname{tr}\left(\eta(X + I)^{-1}(\grave{U} + I)D(\grave{U} + I)^T(X + I)^{-T}\eta^T\right)$$
$$= \operatorname{tr}\left(\eta BDB^T\eta^T\right). \tag{9}$$

With the LDL assignment of $U$, we further have that,

$$\operatorname{tr}\left(\eta BDB^T\eta^T\right) = \operatorname{tr}\left(\eta D\eta^T\right). \tag{10}$$

First, consider the worst-case loss, $\mathcal{L}_{\text{worst}}$. The goal is to construct a particularly bad case where the entries of $\tilde{W}$ are $1/2 \pm \epsilon$, and thus when rounding to the integers we will always have error $1/2$. Construct a weight matrix $\tilde{W} \in \mathbb{R}^{m \times n}$ such that each entry satisfies,

$$\tilde{W}_{ij} = \begin{cases} 0.5 - \epsilon & \text{w.p. } 1/2 \\ 0.5 + \epsilon & \text{w.p. } 1/2 \end{cases} \Rightarrow \eta_{ij} = \begin{cases} +0.5 & \text{w.p. } 1/2 \\ -0.5 & \text{w.p. } 1/2 \end{cases},$$

and the quantization errors $\eta \in \mathbb{R}^{m \times n}$ are for each entry $\{+1/2, -1/2\}$ with equal probability. For this particular $\tilde{W}$, $\mathcal{A}$ achieves proxy loss $\mathcal{L}_{\text{worst}}(\mathcal{A}, H) \overset{(9)}{=} \mathbf{E}\left[\operatorname{tr}\left(\eta BDB^T\eta^T\right)\right] = \frac{m}{4}\operatorname{tr}\left(BDB^T\right)$, with $\mathcal{Q}$ as either nearest or stochastic rounding. It follows from the supremum in the definition of $\mathcal{L}_{\text{worst}}$ in Eq. (5) that, $\mathcal{L}_{\text{worst}}(\mathcal{A}, H) \geq \frac{m}{4}\operatorname{tr}\left(BDB^T\right)$. For the LDL assignment of $U$, the worst case expected quantization error rounding to the integers is $1/2$. Therefore, $\mathcal{L}_{\text{worst}}(\text{LDLQ}, H) \overset{(10)}{=} \frac{m}{4}\operatorname{tr}\left(D\right)$, again for $\mathcal{Q}$ as either nearest or stochastic rounding. $B$ must be a unit triangular matrix since it is the product of unit triangular matrices. Therefore $\operatorname{tr}\left(BDB^T\right)$ is minimized when $B = I$, and

$$\mathcal{L}_{\text{worst}}(\text{LDLQ}, H) \leq \mathcal{L}_{\text{worst}}(\mathcal{A}, H).$$

Next, consider the average loss, $\mathcal{L}_{\text{avg}}$, where $W \sim Unif[0, 1]^{m \times n}$. For $\mathcal{Q}$ as nearest rounding, the entries of the quantization error $\eta$ are $Unif[-\frac{1}{2}, \frac{1}{2}]$, because each entry is independent and uniformly distributed. It follows that for any entry of $\eta$, $\mathbf{E}\left[\eta_{ij}^2\right] = \int_{-1/2}^{1/2} x^2 dx = \frac{1}{12}$. Therefore, $\mathcal{L}_{\text{avg}}(\mathcal{A}, H) \overset{(9)}{=} \mathbf{E}_{W \sim Unif[0,1]^{m \times n}}\left[\operatorname{tr}\left(\eta BDB^T\eta^T\right)\right] = \frac{m}{12}\operatorname{tr}\left(BDB^T\right)$. For $\mathcal{Q}$ as stochastic rounding, the entries of the quantization error $\eta$ are $Unif[-1, 1]$. It follows that for any entry of $\eta$, $\mathbf{E}\left[\eta_{ij}^2\right] = \int_0^1 x(1 - x)dx = \frac{1}{6}$. Note that for stochastic rounding, the quantization error will be $x$ with probability $(1 - |x|)$. Therefore, $\mathcal{L}_{\text{avg}}(\mathcal{A}, H) = \frac{m}{6}\operatorname{tr}\left(BDB^T\right)$. Based on these same calculations of $\mathbf{E}\left[\eta_{ij}^2\right]$,

we have that $\mathcal{L}_{\mathrm{avg}}(LDL, H) \stackrel{(9)}{=} \frac{m}{12} \operatorname{tr}(D)$ with $\mathcal{Q}$ as nearest , and $= \frac{m}{6} \operatorname{tr}(D)$ with $\mathcal{Q}$ as stochastic rounding. By the same reasoning on the minimization of $\operatorname{tr}(BDB^T)$,

$$\mathcal{L}_{\mathrm{avg}}(\mathsf{LDLQ}, H) \leq \mathcal{L}_{\mathrm{avg}}(\mathcal{A}, H).$$

$\square$

**Subsection 3.3 (Incoherence: Optimality with a Spectral Bound)**

**Definition 1.** *We say a symmetric Hessian matrix $H \in \mathbb{R}^{n \times n}$ is $\mu$-incoherent if it has an eigendecomposition $H = Q\Lambda Q^T$ such that for all $i$ and $j$, $|Q_{ij}| = |e_i^T Q e_j| \leq \mu/\sqrt{n}$. By extension, we say a weight matrix $W \in \mathbb{R}^{m \times n}$ is $\mu$-incoherent if all $i$ and $j$, $|W_{ij}| = |e_i^T W e_j| \leq \mu \|W\|_F /\sqrt{mn}$.*

**Lemma 8.** *Let $H \in \mathbb{R}^{n \times n}$ be a positive semi-definite symmetric matrix, and let $a_1, \ldots, a_n$ be a sequence of vectors in $\mathbb{R}^n$. Consider the recurrence given by $\Sigma_0 = 0 \in \mathbb{R}^{n \times n}$ and from $k = 0$ to $n - 1$*

$$\Sigma_{k+1} = (I - e_k a_k^T)\Sigma_k(I - a_k e_k^T) + e_k e_k^T.$$

*Let $\ell(a_1, \ldots, a_n) = \operatorname{tr}(H\Sigma_n)$. Then if $H = LDL^T$ is the LDL decomposition of H, a global minimum of $\ell$ occurs when $a_k$ is the kth column of L, and at this minimum, $\ell = \operatorname{tr}(D)$.*

*Proof.* First observe that at step $k$, $\Sigma_k$ will be 0 in all entries $(\Sigma_k)_{ij}$ if $\min(i, j) \geq k$. This means that changing the last $n - k$ entries of $a_k$ does not change $\Sigma$ (or $\ell$) at all. Without loss of generality, set those entries of $a_k$ to 0. If $A$ is the matrix whose $k$th row is $a_k$, this is equivalent to saying that $A$ is strictly lower triangular.

Next, let $\eta$ be a random Gaussian sampled from $\mathcal{N}(0, I)$, and consider the recurrence given by $x_0 = 0 \in \mathbb{R}^n$ and

$$x_{k+1} = x_k - e_k a_k^T x_k + e_k e_k^T \eta.$$

It's straightforward to see that $\Sigma_k = \mathbf{E}[x_k x_k^T]$. But it's also easy to see that the step-$k$ update only modifies/assigns the $k$th entry of $x$, and does so based only on earlier entries of $x$. Since $e_k^T x_k = 0$, and no later step assigns the $k$-or-lower entries of $x$,

$$e_k^T x_n = e_k^T x_{k+1} = 0 - a_k^T x_k + e_k^T \eta = -a_k^T x_n + e_k^T \eta,$$

which in vector form yields

$$(I + A)x_n = \eta.$$

In particular, this immediately implies that

$$\Sigma_n = (I + A)^{-1}(I + A)^{-T}$$

and

$$\ell = \operatorname{tr}(H\Sigma_n) = \operatorname{tr}\left((I + A)^{-T}H(I + A)^{-1}\right) = \operatorname{tr}\left(B^{-T}HB^{-1}\right).$$

where $B = I + A$. Differentiating with respect to $B$ in strictly lower triangular direction $\Delta$ (the only direction in which we have degress of freedom, since the diagonal of $B$ must be unit) yields

$$-2\operatorname{tr}\left(B^{-T}HB^{-1}\Delta B^{-1}\right).$$

It's not hard to see that if $H = LDL^T$ is the LDL decomposition of $H$, and $B^T = L$, that the gradient is

$$-2\operatorname{tr}\left(D\Delta B^{-1}\right) = -2\operatorname{tr}\left(\Delta B^{-1}D\right) = -2\langle\Delta^T, B^{-1}D\rangle.$$

Since $\Delta^T$ is strictly upper triangular, but $B^{-1}D$ must be lower triangular, this is 0 so we have a minimum. The uniqueness of this minimum (up to assignments of the lower-triangular elements of $A$ or $B$, which have no effect on $\ell$) also immediately follows from the recurrence relation. This implies the minimum is global. This is what we wanted to show. $\square$

**Lemma 2.** *Let $H \in \mathbb{R}^{n \times n}$ be a $\mu$-incoherent positive semi-definite symmetric matrix and let $H = (\dot{U} + I)D(\dot{U} + I)^T$ be its LDL Cholesky decomposition, where $\dot{U}$ is a strictly upper triangular matrix and $D$ is a (non-negative) diagonal matrix. Then,*

$$\operatorname{tr}(D) \leq \frac{\mu^2}{n}\operatorname{tr}\left(H^{1/2}\right)^2.$$

*Proof.* By continuity of $\operatorname{tr}(D)$ and $\operatorname{tr}\left(H^{1/2}\right)$, it suffices to prove the lemma for positive definite $H$. First, the closure of positive definite symmetric matrices is the set of positive semi-definite symmetric matrices. Second, consider the set of $H$ that are positive definite and satisfy $\frac{\mu^2}{n}\operatorname{tr}\left(H^{1/2}\right)^2 - \operatorname{tr}(D) \geq 0$, i.e. are non-negative. The closure of this set (i.e. $H \succeq 0$) must also satisfy that the inequality is non-negative.

Let $H = Q\Lambda Q^T$ be the eigendecomposition of $H$. First, observe that by incoherence,

$$e_k^T H^{1/2} e_k = \sum_{i=1}^n \lambda_i^{1/2}(e_i^T Q e_k)^2 \leq \frac{\mu^2}{n}\sum_{i=1}^n \lambda_i^{1/2} = \frac{\mu^2}{n}\operatorname{tr}\left(H^{1/2}\right).$$

Set

$$\alpha = \frac{\mu^2}{n}\operatorname{tr}\left(H^{1/2}\right),$$

and consider the recurrence from Lemma 8 with

$$a_k = \frac{H^{1/2}e_k}{\alpha}$$

Then

$$\Sigma_{k+1} = \left(I - \alpha^{-1}e_k e_k^T H^{1/2}\right)\Sigma_k\left(I - \alpha^{-1}H^{1/2}e_k e_k^T\right) + e_k e_k^T.$$

Suppose by way of induction that for some scalar the covariance $\Sigma_k \preceq \alpha H^{-1/2}$. For the base case, this obviously holds since $\Sigma_0 = 0$. At step $k$,

$$\Sigma_{k+1} \preceq \left(I - \alpha^{-1}e_k e_k^T H^{1/2}\right)\alpha H^{-1/2}\left(I - \alpha^{-1}H^{1/2}e_k e_k^T\right) + e_k e_k^T$$

$$= \alpha H^{-1/2} - 2e_k e_k^T + \alpha^{-1}e_k e_k^T H^{1/2}e_k e_k^T + e_k e_k^T$$

$$\preceq \alpha H^{-1/2}.$$

Note that with this assignment,

$$a_k^T \Sigma_k a_k \leq (\alpha^{-1}e_k^T H^{1/2})(\alpha H^{-1/2})(\alpha^{-1}H^{1/2}e_k) = \alpha^{-1}e_k^T H^{1/2}e_k \leq 1.$$

So, by induction it follows that

$$\Sigma_n \preceq \frac{\mu^2}{n}\operatorname{tr}\left(H^{1/2}\right)\cdot H^{-1/2},$$

and so

$$\operatorname{tr}(H\Sigma_n) \leq \frac{\mu^2}{n}\operatorname{tr}\left(H^{1/2}\right)\operatorname{tr}\left(H\cdot H^{-1/2}\right) = \frac{\mu^2}{n}\operatorname{tr}\left(H^{1/2}\right)^2.$$

But from Lemma 8, we know that $\operatorname{tr}(D)$ is the global minimum of $\operatorname{tr}(H\Sigma_n)$ for any assignment of $a_k$. This immediately gives us the desired result. $\qquad\square$

**Lemma 3.** *Let $H$ be symmetric positive definite. In the worst case stochastic rounding achieves $\mathcal{L}_{\mathrm{worst}}(\mathsf{Stoch}, H) = (m/4)\operatorname{tr}(H)$. In the average case nearest and stochastic rounding achieve $\mathcal{L}_{\mathrm{avg}}(\{\mathsf{Near}, \mathsf{Stoch}\}, H) = (m/c)\operatorname{tr}(H)$, where $c = 12$ for nearest, and $c = 6$ for stochastic.*

*Proof.* For nearest and stochastic rounding, set the linear feedback $U$ in Eq. (2) to be zero. Stochastic rounding achieves worst-case loss,

$$\mathcal{L}_{\mathrm{worst}}(\mathsf{Stoch}, H) \overset{(3)}{=} \sup_{W \in \mathbb{R}^{m \times n}} \mathbf{E}\left[\operatorname{tr}\left(\eta H \eta^T\right)\right] = \frac{m}{4}\operatorname{tr}(H). \qquad (11)$$

For the average-case proxy loss, recall the computations of $\mathbf{E}\left[\eta_{ij}^2\right]$ from the proof of Theorem 1.

$$\mathcal{L}_{\mathrm{avg}}(\mathsf{Near}, H) \overset{(3)}{=} \mathbf{E}_{W \sim Unif[0,1]^{m \times n}}\left[\operatorname{tr}\left(\eta H \eta^T\right)\right] = \frac{m}{12}\operatorname{tr}(H) \qquad (12)$$

$$\mathcal{L}_{\mathrm{avg}}(\mathsf{Stoch}, H) \overset{(3)}{=} \mathbf{E}_{W \sim Unif[0,1]^{m \times n}}\left[\operatorname{tr}\left(\eta H \eta^T\right)\right] = \frac{m}{6}\operatorname{tr}(H). \qquad (13)$$

$$\square$$

**Without incoherence: no improvement with a spectral bound**

**Theorem 4.** *Consider all $\tilde{H}$ with the same spectrum as $H$. For any positive semi-definite $H$, the following holds. On the worst-case loss* LDLQ *achieves the same error as stochastic rounding,*

$$\sup_{\tilde{H} s.t. \, \mathrm{eig}(\tilde{H})=\mathrm{eig}(H)} \mathcal{L}_{\mathrm{worst}}(\mathsf{LDLQ}, \tilde{H}) = \mathcal{L}_{\mathrm{worst}}(\mathsf{Stoch}, H) = \frac{m}{4} \mathrm{tr}\,(H) \,.$$

*On the average-case loss* LDLQ *achieves the same error as the corresponding rounding routine. Let* $\mathcal{B} = \{\mathsf{Near}, \mathsf{Stoch}\}$ *and* $c = 12$ *for nearest,* $c = 6$ *for stochastic.*

$$\sup_{\tilde{H} s.t. \, \mathrm{eig}(\tilde{H})=\mathrm{eig}(H)} \mathcal{L}_{\mathrm{avg}}(\mathsf{LDLQ}^*, \tilde{H}) = \mathcal{L}_{\mathrm{avg}}(\mathcal{B}, H) = \frac{m}{c} \mathrm{tr}\,(H) \,.$$

*Proof.* See Lemma 3 for calculations on the proxy loss for nearest and stochastic rounding.

For LDLQ, we will derive lower and upper bounds on $\sup_{\tilde{H} s.t. \, \mathrm{eig}(\tilde{H})=\mathrm{eig}(H)} \mathcal{L}_{\mathrm{worst}}(\mathsf{LDLQ}, \tilde{H})$ and $\sup_{\tilde{H} s.t. \, \mathrm{eig}(\tilde{H})=\mathrm{eig}(H)} \mathcal{L}_{\mathrm{avg}}(\mathsf{LDLQ}, \tilde{H})$, and show they are equal. To construct a lower bound, consider $\tilde{H} = I\Lambda I$ where $\Lambda$ are the eigenvalues of $H$. This decomposition is also the LDL decomposition of $\tilde{H}$, rewritten as $\tilde{H} = (U + I)D(U + I)^{-1}$. It follows that $\mathrm{tr}\,(D) = \mathrm{tr}\left(\tilde{H}\right)$ for this $\tilde{H}$. Combine this result with the worst and average-case losses calculated in the proof of Theorem 1. For the worst-case loss from the proof of Theorem 1, $\geq \frac{m}{4} \mathrm{tr}\,(H)$. The lower bound for the average-case loss is $\geq \frac{m}{12} \mathrm{tr}\,(H)$ for $\mathcal{Q}$ as nearest, and $\geq \frac{m}{6} \mathrm{tr}\,(H)$ for $\mathcal{Q}$ as stochastic. Now upper bounds are derived using the preceding calculations in Eq. (11)-(13), and using the worst and average-case optimality of LDLQ proven in Theorem 1. The lower and upper bounds are tight, proving our result. $\square$

## E   Proofs for Section 4 (Quantization With Incoherence Processing: Incoherence Processing Step )

**Subsection 4.1 (Incoherence via Efficient Orthogonal Multiplication)**

**Lemma 9** (Theorem 2.4 from Lalley [2] )**.** *There exist constants $C$ and $A$ independent of $n$ such that for any function $F$ from the unit sphere in $n$ dimensions to $\mathbb{R}$ that is 1-Lipschitz relative to the Riemannian metric on the sphere,*

$$\mathbf{P}_{x \sim \mathcal{S}_n} \left( F(x) - \mathbf{E}_{x \sim \mathcal{S}_n}[F(x)] \geq t \right) \leq C \exp\left( -\frac{nt^2}{A} \right)$$

**Lemma 10.** *Let $B \in \mathbb{R}^{m \times n}$ be a matrix, and let $x$ be a random vector uniformly distributed on the unit sphere in $\mathbb{R}^n$. Then there exist global constants $A > 0$ and $C > 0$ independent of $m$ and $n$ such that*

$$\mathbf{P} \left( \|Bx\|^2 \geq \frac{A \|B\|_F^2}{n} \log\left( \frac{C}{\delta} \right) \right) \leq \delta,$$

*Proof.* Let

$$F(x) = \frac{\|Bx\|}{\|B\|_F}.$$

Observe that

$$\nabla F(x) = \frac{B^T Bx}{\|Bx\| \cdot \|B\|_F},$$

and so

$$\|\nabla F(x)\| \leq 1.$$

Also observe that for $x$ drawn uniformly from the sphere in $n$ dimensions,

$$\mathbf{E}\,[F(x)] \leq \sqrt{\mathbf{E}\,[F(x)^2]} = \frac{1}{\|B\|_F} \cdot \sqrt{\mathbf{E}\left[ \|Bx\|^2 \right]} = \frac{1}{\sqrt{n}}.$$

So, applying Lemma 9,

$$\mathbf{P}\left(\frac{\|Bx\|}{\|B\|_F} - \frac{1}{\sqrt{n}} \geq t\right) \leq C\exp\left(-\frac{nt^2}{A}\right).$$

If we let $\delta$ be

$$\delta = C\exp\left(-\frac{nt^2}{A}\right),$$

then

$$\frac{A}{n}\log\left(\frac{C}{\delta}\right) = t^2$$

Trivially, then, for some modified global constants $A'$ and $C'$,

$$\frac{A'}{n}\log\left(\frac{C'}{\delta}\right) = \left(t + \frac{1}{\sqrt{n}}\right)^2$$

This means that

$$\mathbf{P}\left(\frac{\|Bx\|^2}{\|B\|_F^2} \geq \frac{A'}{n}\log\left(\frac{C'}{\delta}\right)\right) \leq \delta,$$

i.e.

$$\mathbf{P}\left(\|Bx\|^2 \geq \frac{A'\|B\|_F^2}{n}\log\left(\frac{C'}{\delta}\right)\right) \leq \delta,$$

This is what we wanted to prove. $\qquad\square$

**Lemma 5.** *Let $H$ be a positive semi-definite matrix on $\mathbb{R}^{n\times n}$ and $W$ a matrix on $\mathbb{R}^{m\times n}$, and suppose that $m = p_1 \cdot p_2 \cdots p_k$ and $n = q_1 \cdot q_2 \cdots q_k$. Let $U_1, U_2, \ldots, U_k, V_1, V_2, \ldots, V_k$ be independent random orthogonal matrices on $\mathbb{R}^{p_i\times p_i}$ and $\mathbb{R}^{q_i\times q_i}$ respectively. Set $U$ as the Kronecker product $U = U_1 \otimes U_2 \otimes \cdots \otimes U_k$ and $V$ as $V = V_1 \otimes V_2 \otimes \cdots \otimes V_k$ Then $VHV^T$ is $\mu_H$-incoherent with probability at least $1 - \delta$, and $UWV^T$ is $\mu_W$-incoherent with probability at least $1 - \delta$, where*

$$\mu_H = A^{k/2}\log\left(\frac{Ckn^2}{\delta}\right)^{k/2} = \tilde{\mathcal{O}}(1) \quad\text{and}\quad \mu_W = A^k\log\left(\frac{2Ckmn}{\delta}\right)^k = \tilde{\mathcal{O}}(1)$$

*for some global constants $A$ and $C$ independent of $n$ and $k$.*

*Proof.* First we will prove what we want to prove about $H$; then we will prove what we want to prove about $W$. Let $Q$ be a matrix of eigenvectors of $H$. Observe that since $Q$ is an orthogonal matrix (by the spectral theorem, because $H$ is symmetric), $Qe_j$ is a unit vector, i.e. $\|Qe_j\| = 1$. Call $Qe_j = y$. Also observe that

$$e_i^T(U_1 \otimes U_2 \otimes \cdots \otimes U_k) = ((e_{i_1}^T U_1) \otimes (e_{i_2}^T U_2) \otimes \cdots \otimes (e_{i_k}^T U_k))$$

for some indices $i_j$. Call $e_{i_j}^T U_j = x_j^T$, and observe that the $x_j$ are all independent unit random vectors. So,

$$((U_1 \otimes U_2 \otimes \cdots \otimes U_k)Q)_{ij} = (x_1 \otimes x_2 \otimes \cdots \otimes x_k)^T y$$

for random unit vectors $x_1, \ldots, x_k$ and unit vector $y$. We can easily bound this with $k$ applications of Lemma 10 and a union bound, yielding

$$\mathbf{P}\left(((x_1 \otimes x_2 \otimes \cdots \otimes x_k)^T y)^2 \geq \frac{A^k}{n}\log\left(\frac{C}{\delta}\right)^k\right) \leq k\delta,$$

Setting $\delta \mapsto \frac{\delta}{kn^2}$ yields

$$\mathbf{P}\left(((x_1 \otimes x_2 \otimes \cdots \otimes x_k)^T y)^2 \geq \frac{A^k}{n}\log\left(\frac{Ckn^2}{\delta}\right)^k\right) \leq \frac{\delta}{n^2},$$

and unioning over all the entries of the large orthogonal matrix,

$$\mathbf{P}\left(\max_{i,j}\left|((U_1 \otimes U_2 \otimes \cdots \otimes U_k)Q)_{ij}\right| \geq \sqrt{\frac{A^k}{n}\log\left(\frac{Ckn^2}{\delta}\right)^k}\right) \leq \delta.$$

Next, for $W$, observe that if we flatten $W$, then $W/\|W\|_F$ is a unit vector. Then any entry of the resulting matrix can be written as

$$(x_1 \otimes x_2 \otimes \cdots \otimes x_k)^T W (y_1 \otimes y_2 \otimes \cdots \otimes y_k)$$

where $x_1, \ldots, x_k$ and $y_1, \ldots, y_k$ are $k$ independent random unit vectors. We can easily bound this with $2k$ applications of Lemma 10 and a union bound, yielding

$$\mathbf{P}\left( \left((x_1 \otimes x_2 \otimes \cdots \otimes x_k)^T W (y_1 \otimes y_2 \otimes \cdots \otimes y_k)\right)^2 \geq \frac{A^{2k}}{mn} \log\left(\frac{C}{\delta}\right)^{2k} \right) \leq 2k\delta,$$

Setting $\delta \mapsto \frac{\delta}{2kmn}$ yields

$$\mathbf{P}\left( \left((x_1 \otimes x_2 \otimes \cdots \otimes x_k)^T W (y_1 \otimes x_2 \otimes \cdots \otimes y_k)\right)^2 \geq \frac{A^{2k}}{mn} \log\left(\frac{2Ckmn}{\delta}\right)^{2k} \right) \leq \frac{\delta}{mn},$$

and unioning over all the $mn$ entries of the large orthogonal matrix,

$$\mathbf{P}\left( \max_{i,j} \left| e_i^T (U_1 \otimes U_2 \otimes \ldots U_k) W (V_1 \otimes V_2 \otimes \cdots \otimes V_k) e_j \right| \geq \sqrt{\frac{A^{2k}}{mn} \log\left(\frac{2Ckmn}{\delta}\right)^{2k}} \right) \leq \delta.$$

This is what we wanted to show.

$\square$

# F  Proofs for Section 5 (Extensions and Further Analyses)

### Subsection 5.1 (OPTQ is a Special Case of LDLQ)

**Theorem 6.** *OTPQ [1] falls within the class of adaptive rounding procedures with linear feedback as described by Eq. (2), and is equivalent to LDLQ in Section 3.*

*Proof.* OPTQ works in the following way. After OPTQ has quantized the first $t - 1$ components of the row vector $w$, it minimizes the proxy loss over the remaining $n - t + 1$ elements, keeping the first $t - 1$ elements fixed. It then quantizes the $t$th element using nearest rounding to the grid and clamping. It then proceeds to the next column. If we let $\Delta = \hat{w} - w$, this proxy loss that it minimizes can be written in block form as

$$\ell = \Delta_{1:(t-1)} H_{1:(t-1),1:(t-1)} \Delta_{1:(t-1)}^T + 2\Delta_{1:(t-1)} H_{1:(t-1),t:n} + \Delta_{t:n} H_{t:n,t:n} \Delta_{t:n}^T$$

and its minimum over $\Delta_{t:n}$ will occur when

$$0 = \Delta_{1:(t-1)} H_{1:(t-1),t:n} + \Delta_{t:n} H_{t:n,t:n},$$

i.e.

$$\Delta_{t:n} = -\Delta_{1:(t-1)} H_{1:(t-1),t:n} \left(H_{t:n,t:n}\right)^{-1}.$$

Now, suppose that $H = \tilde{U} D \tilde{U}^T$ is the LDL decomposition of $H$, where $\tilde{U}$ is unit upper triangular and $D$ is diagonal. Since $\tilde{U}$ is upper triangular,

$$H_{t:n,t:n} = \tilde{U}_{t:n,t:n} D_{t:n,t:n} \tilde{U}_{t:n,t:n}^T.$$

Similarly,

$$H_{1:(t-1),t:n} = \tilde{U}_{1:(t-1),t:n} D_{t:n,t:n} \tilde{U}_{t:n,t:n}^T.$$

This means that

$$\Delta_{t:n} = -\Delta_{1:(t-1)} \tilde{U}_{1:(t-1),t:n} \left(\tilde{U}_{t:n,t:n}\right)^{-1}.$$

Now, the only part of the value of $\Delta_{t:n}$ which matters is the first entry, since this is the one that's going to be used to make the next quantization decision. But since $\tilde{U}_{t:n,t:n}$ is unit upper triangular and so is its inverse, $\left(\tilde{U}_{t:n,t:n}\right)^{-1} e_t = e_t$, and so

$$\Delta_t = \Delta_{t:n} e_1 = -\Delta_{1:(t-1)} \tilde{U}_{1:(t-1),t:n} e_t = -\Delta_{1:(t-1)} \tilde{U}_{1:(t-1),t} = -\Delta(\tilde{U} - I)e_t.$$

Finally, we quantize the $t$-th weight as

$$\hat{w}_t = \mathcal{Q}(w_t - (\hat{W} - W)(\tilde{U} - I)e_t).$$

This update is equivalent to our adaptive rounding with linear feedback procedure in Eq. (2), with $U$ assigned from the LDL decomposition of $H$. $\square$

**Subsection 5.2 (A Bound for Rounding to a Finite Grid)**

Algorithm 5 presents a quantization procedure which theoretically address OPTQ's clamping issue, by incorporating a restriction of $|\hat{W}_{ij} - W_{ij}|$ into objective (7). Note that for simplicity, here we present the explicit case where only two factors are used in each Kronecker product of orthogonal matrices; however, the proof should generalize to any number of factors.

---

**Algorithm 5** "Fixed" Rounding via a Convex Program

---

**Require:** $W \in \mathbb{R}^{m \times n}$, $H \in \mathbb{R}^{n \times n}$, $c > 0$, $\rho > 0$
**Require:** factorization $m = p_1 p_2$, $n = p_3 p_4$
  **draw** $U_1 \in \mathbb{R}^{p_1 \times p_1}$ uniformly from the set of orthogonal matrices using seed seed$(U_1)$
  **draw** $U_2 \in \mathbb{R}^{p_2 \times p_2}$ uniformly from the set of orthogonal matrices using seed seed$(U_2)$
  **draw** $U_3 \in \mathbb{R}^{p_3 \times p_3}$ uniformly from the set of orthogonal matrices using seed seed$(U_3)$
  **draw** $U_4 \in \mathbb{R}^{p_4 \times p_4}$ uniformly from the set of orthogonal matrices using seed seed$(U_4)$
  $W \leftarrow (U_1 \otimes U_2)W(U_3 \otimes U_4)$
  $H \leftarrow (U_3^T \otimes U_4^T)H(U_3 \otimes U_4)$
  $W \leftarrow \frac{2^b - 1}{2}\left(\frac{W}{\rho} + 1\right)$ elementwise
  $W \leftarrow \mathrm{clamp}(W, \min = 0, \max = 2^b - 1))$ elementwise
  use ADMM or some other solver to solve

$$\text{minimize: } \mathrm{tr}\left(HL^TL\right)$$
$$\text{over: } L \text{ unit upper triangular}$$
$$\text{subject to: } e_i^T L^T L e_i \leq 1 + c, \ \forall i \in \{1, \ldots, n\}.$$

  note that when $c = \infty$, $L^{-1}$ is the factor from the LDL decomposition of $H$
  $\dot{U} \leftarrow L^{-1} - I$
  **for** $k \in \{1, \ldots, n\}$ **do** $\hat{W}_k \leftarrow \mathrm{clamp}(\mathcal{Q}(W_k + (W - \hat{W})\dot{U}_k), 0, 2^b - 1)$ ▷ round with LF
  $\hat{W} \leftarrow \rho\left(\frac{2\hat{W}}{2^b - 1} - 1\right)$
  $\hat{W} \leftarrow (U_1^T \otimes U_2^T)\hat{W}(U_3^T \otimes U_4^T)$
  **return** $\hat{W}$ encoded as a tuple of the integer rounded values, the scale factor $\rho$, and the seeds

---

**Lemma 11.** *Suppose that for positive definite $\mu$-incoherent matrix $H \in \mathbb{R}^{n \times n}$ and scalar $c > 0$, $L$ is the solution to the optimization problem*

$$\text{minimize: } \mathrm{tr}\left(HL^TL\right)$$
$$\text{over: } L \text{ unit upper triangular}$$
$$\text{subject to: } e_i^T L^T L e_i \leq 1 + c, \ \forall i \in \{1, \ldots, n\}.$$

*Then the solution satisfies*

$$\mathrm{tr}\left(HL^TL\right) = \frac{\mu^2}{n \cdot \min(1, c)}\mathrm{tr}\left(H^{1/2}\right)^2.$$

*Proof.* Let $\eta \in \mathbb{R}^{1 \times n}$ be a random standard Gaussian variable as a row vector, let $A$ be a matrix, and consider the recurrence relation over $x_t \in \mathbb{R}^{1 \times n}$ given by $x_0 = 0$ and

$$x_t = x_{t-1} - x_{t-1}Ae_ie_i^T + \eta e_i e_i^T$$

We first note that since $x_t$ is supported only on $\{1, \ldots, t\}$, if $M$ denotes the strictly upper triangular mask, this update step is equivalent to

$$x_t = x_{t-1} - x_{t-1}(A \odot M)e_i e_i^T + \eta e_i e_i^T.$$

From here, it's fairly easy to see by induction that

$$x_n = -x_n(A \odot M) + \eta,$$

and so

$$x_n(I + A \odot M) = \eta,$$

or
$$x_n = \eta(I + A \odot M)^{-1}.$$

Now, since $I + A \odot M$ is a unit upper triangular matrix, its inverse is also a unit upper triangular matrix. If we let $L = (I + A \odot M)^{-1}$, then $L$ is a unit upper triangular matrix and

$$\mathbf{E}\left[x_n^T x_n\right] = L^T L.$$

We are going to choose $A$ such that $L$ is a feasible solution to our optimization problem and has the desired objective. Next, let $\Sigma_t = \mathbf{E}\left[x_t^T x_t\right]$, and observe that

$$\Sigma_t = \left(I - Ae_i e_i^T\right)^T \Sigma_{t-1}\left(I - Ae_i e_i^T\right) + e_i e_i^T.$$

Let $\alpha > 0$ be some constant to be set later, and set $A = \alpha H^{1/2}$. Suppose by way of induction that for some constant $\beta > 0$ to be set later, $\Sigma_t \preceq \beta H^{-1/2}$. The base case clearly holds since $\Sigma_0 = 0$. For the inductive step,

$$\Sigma_t \preceq \beta \left(I - \alpha H^{1/2} e_i e_i^T\right)^T H^{-1/2}\left(I - \alpha H^{1/2} e_i e_i^T\right) + e_i e_i^T$$
$$= \beta H^{-1/2} - 2\alpha\beta e_i e_i^T + \alpha^2 \beta e_i e_i^T H^{1/2} e_i e_i^T + e_i e_i^T.$$

This inductive step will hold if, letting $h = \max_i e_i^T H^{1/2} e_i$,

$$2\alpha\beta \geq 1 + \alpha^2 \beta h$$

On the other hand,

$$e_i^T L^T L e_i = \mathbf{E}\left[(x_n e_i)^2\right]$$
$$= \mathbf{E}\left[(-x_{i-1}Ae_i + \eta e_i)^2\right]$$
$$= \mathbf{E}\left[(-x_{i-1}Ae_i)^2\right] + 1$$
$$= e_i^T A^T \Sigma_{i-1} A e_i + 1$$
$$= \alpha^2 e_i^T H^{1/2} \Sigma_{i-1} H^{1/2} e_i + 1$$
$$\leq \alpha^2 \beta e_i^T H^{1/2} H^{-1/2} H^{1/2} e_i + 1$$
$$\leq \alpha^2 \beta e_i^T H^{1/2} e_i + 1.$$

So the constraint of our optimization problem will be satisfied if

$$\alpha^2 \beta h \leq c.$$

To satisfy these constraints, set $\beta = \max(h, h/c)$ and $\alpha = \beta^{-1}$. Then

$$2\max(h, h/c)^{-1} \cdot \max(h, h/c) \geq 1 + \max(h, h/c)^{-2} \cdot \max(h, h/c) \cdot h,$$

and

$$\max(h, h/c)^{-2} \cdot \max(h, h/c) \cdot h \leq c.$$

Also, the objective will be bounded by

$$\operatorname{tr}\left(HL^T L\right) = \operatorname{tr}\left(H\Sigma_n\right) \leq \beta \operatorname{tr}\left(H^{1/2}\right) = \max(1, c^{-1}) \cdot h \cdot \operatorname{tr}\left(H^{1/2}\right).$$

Now, applying incoherence to bound $h$, where $H = U\Lambda U^T$ is the eigendecomposition of $H$,

$$e_i^T H^{1/2} e_i = \sum_{j=1}^n \lambda_j^{1/2}(e_i^T U e_j)^2 \leq \sum_{j=1}^n \lambda_j^{1/2}\frac{\mu^2}{n} = \frac{\mu^2}{n}\operatorname{tr}\left(H^{1/2}\right).$$

So this yields a whole bound of

$$\operatorname{tr}\left(HL^T L\right) = \frac{\mu^2}{n \cdot \min(1, c)}\operatorname{tr}\left(H^{1/2}\right)^2.$$

This is what we wanted to show. $\qquad\square$

**Lemma 12.** *Suppose that we quantize the row vector $w \in \mathbb{R}^{1 \times n}$ using $L$ the solution to the optimization problem*

$$\text{minimize: } \text{tr}\left(HL^T L\right)$$
$$\text{over: } L \text{ unit upper triangular}$$
$$\text{subject to: } e_i^T L^T L e_i \leq 1 + c, \ \forall i \in \{1, \ldots, n\}$$

*and*

$$\hat{w} = \mathcal{Q}_{\text{stoch}}\left(w - (\hat{w} - w)(L^{-1} - I)\right),$$

*where $\mathcal{Q}_{\text{stoch}}$ denotes elementwise unbiased stochastic rounding. Then for any $u \in \mathbb{R}^n$ and any $\delta > 0$*

$$\mathbf{P}\left(|(\hat{w} - w)u| \geq \|Lu\| \sqrt{\frac{1}{2} \log\left(\frac{2}{\delta}\right)}\right) \leq \delta.$$

*In particular,*

$$\mathbf{P}\left(|(\hat{w} - w)(L^{-1} - I)e_i| \geq \sqrt{\frac{c}{2} \log\left(\frac{2}{\delta}\right)}\right) \leq \delta.$$

*Proof.* Let $\eta$ be the error of stochastic rounding, and observe that each entry is, conditioned on earlier steps, zero mean and supported on two values that differ by $1$. Also observe that

$$\hat{w} = \left(w - (\hat{w} - w)(L^{-1} - I)\right) + \eta,$$

and so

$$\hat{w} - w = \eta L$$

and

$$\mathbf{E}\left[\exp\left((\hat{w} - w)u\right)\right] = \mathbf{E}\left[\exp\left(\eta L u\right)\right].$$

From a repeated application of Hoeffding's lemma, we get

$$\mathbf{E}\left[\exp\left((\hat{w} - w)u\right)\right] \leq \exp\left(\frac{1}{8} \|Lu\|^2\right).$$

Setting $u \mapsto \gamma u$ for $\gamma > 0$,

$$\mathbf{E}\left[\exp\left(\gamma(\hat{w} - w)u\right)\right] \leq \exp\left(\frac{\gamma^2}{8} \|Lu\|^2\right).$$

And by Markov's inequality,

$$\mathbf{P}\left(\exp\left(\gamma(\hat{w} - w)u\right) \geq \exp(\gamma R)\right) \leq \exp(-\gamma R) \exp\left(\frac{\gamma^2}{8} \|Lu\|^2\right),$$

i.e.

$$\mathbf{P}\left((\hat{w} - w)u \geq R\right) \leq \exp\left(-\gamma R + \frac{\gamma^2}{8} \|Lu\|^2\right).$$

Minimizing the right side over $\gamma$ yields $\gamma = 4R \|Lu\|^{-2}$ and

$$\mathbf{P}\left((\hat{w} - w)u \geq R\right) \leq \exp\left(-2R^2 \|Lu\|^{-2}\right).$$

By a union bound,

$$\mathbf{P}\left(|(\hat{w} - w)u| \geq R\right) \leq 2 \exp\left(-2R^2 \|Lu\|^{-2}\right).$$

Now setting the right side equal to $\delta$,

$$\mathbf{P}\left(|(\hat{w} - w)u| \geq \|Lu\| \sqrt{\frac{1}{2} \log\left(\frac{2}{\delta}\right)}\right) \leq \delta.$$

This is what we wanted to show. The second statement follows from the fact that

$$\left\|L(L^{-1} - I)e_i\right\|^2 = \|e_i - Le_i\|^2 = e_i^T e_i - e_i^T L e_i - e_i^T L^T e_i + e_i^T L^T L e_i \leq 1 - 1 - 1 + (1 + c) = c.$$

$\square$

**Lemma 13.** *Suppose that we quantize the row vector* $w \in \mathbb{R}^{1 \times n}$ *using* $L$ *the solution to the optimization problem*

$$\text{minimize: } \operatorname{tr}\left(HL^T L\right)$$
$$\text{over: } L \text{ unit upper triangular}$$
$$\text{subject to: } e_i^T L^T L e_i \leq 1 + c, \ \forall i \in \{1, \ldots, n\}$$

*and*

$$\hat{w} = \mathcal{Q}_{\text{stoch}}\left(w - (\hat{w} - w)(L^{-1} - I)\right),$$

*where* $\mathcal{Q}_{\text{stoch}}$ *denotes elementwise unbiased stochastic rounding. Suppose that for some integer* $b$, $1 \leq w_{ij} \leq 2^b - 2$. *Then if we set*

$$c = 2 \left(\log\left(\frac{4mn}{\delta}\right)\right)^{-1},$$

*then with probability at least* $1 - \delta$, $0 \leq \hat{w}_{ij} \leq 2^b - 1$ *and*

$$\operatorname{tr}\left((\hat{w} - w)H(\hat{w} - w)^T\right) \leq \frac{\mu^2 m}{4n} \operatorname{tr}\left(H^{1/2}\right)^2 \left(\log\left(\frac{4mn}{\delta}\right)^2\right).$$

*Proof.* First, from the previous lemmas, if $Ue_i$ is the $i$th eigenvector of $H$, with eigenvalue $\lambda_i$ since

$$\mathbf{P}\left(\lambda_i(e_j^T(\hat{w} - w)Ue_i)^2 \geq \lambda_i \|LUe_i\|^2 \cdot \frac{1}{2}\log\left(\frac{2}{\delta}\right)\right) \leq \delta.$$

By the union bound,

$$\mathbf{P}\left(\exists i, j, \ \lambda_i(e_j^T(\hat{w} - w)Ue_i)^2 \geq \lambda_i \|LUe_i\|^2 \cdot \frac{1}{2}\log\left(\frac{2mn}{\delta}\right)\right) \leq \delta.$$

And so

$$\mathbf{P}\left(\sum_{i,j} \lambda_i(e_j^T(\hat{w} - w)Ue_i)^2 \geq \sum_{i,j} \lambda_i \|LUe_i\|^2 \cdot \frac{1}{2}\log\left(\frac{2mn}{\delta}\right)\right) \leq \delta,$$

which simplifies to

$$\mathbf{P}\left(\operatorname{tr}\left((\hat{w} - w)H(\hat{w} - w)^T\right) \geq m\operatorname{tr}\left(HL^T L\right) \cdot \frac{1}{2}\log\left(\frac{2mn}{\delta}\right)\right) \leq \delta.$$

Now applying the other lemma,

$$\mathbf{P}\left(\operatorname{tr}\left((\hat{w} - w)H(\hat{w} - w)^T\right) \geq \frac{\mu^2 m}{2n \cdot \min(1, c)} \operatorname{tr}\left(H^{1/2}\right)^2 \log\left(\frac{2mn}{\delta}\right)\right) \leq \delta.$$

And substituting $\delta \mapsto \delta/2$,

$$\mathbf{P}\left(\operatorname{tr}\left((\hat{w} - w)H(\hat{w} - w)^T\right) \geq \frac{\mu^2 m}{2n \cdot \min(1, c)} \operatorname{tr}\left(H^{1/2}\right)^2 \log\left(\frac{4mn}{\delta}\right)\right) \leq \frac{\delta}{2}.$$

On the other hand, again by a union bound from the previous lemma,

$$\mathbf{P}\left(\exists i, j, \ |e_j^T(\hat{w} - w)(L^{-1} - I)e_i| \geq \sqrt{\frac{c}{2}\log\left(\frac{4mn}{\delta}\right)}\right) \leq \frac{\delta}{2}.$$

Setting

$$c = 2 \left(\log\left(\frac{4mn}{\delta}\right)\right)^{-1}$$

yields

$$\mathbf{P}\left(\exists i, j, \ |e_j^T(\hat{w} - w)(L^{-1} - I)e_i| \geq 1\right) \leq \frac{\delta}{2}.$$

And so by another union bound, the probability that

$$\text{tr}\left((\hat{w} - w)H(\hat{w} - w)^T\right) \le \frac{\mu^2 m}{4n} \text{tr}\left(H^{1/2}\right)^2 \left(\log\left(\frac{4mn}{\delta}\right)\right)^2$$

and

$$\max_{i,j} \left|e_j^T(\hat{w} - w)(L^{-1} - I)e_i\right| \le 1$$

is no less than $1 - \delta$. It's clear that if this second inequality holds, the value we pass in to the stochastic quantizer will be in range, and thus so will the output. This proves what we want. $\square$

**Theorem 14.** *Suppose that we are given an input matrix $w$ with bounded maximum entry magnitude $\|w\|_\infty$ and we want to quantize it using $b$ bits. Suppose that we first re-scale the entries of $w$ by mapping*

$$w_{ij} \mapsto \frac{2^b - 3}{2}\left(\frac{w_{ij}}{\|w\|_\infty} + 1\right) + 1;$$

*this guarantees that $1 \le w_{ij} \le 2^b - 2$. Then, suppose we quantize using the procedure described in the previous lemma. Finally, we undo the scaling. Then then with probability at least $1 - \delta$, all the quantized weights will be in range (no overflow or need for clipping) and*

$$\text{tr}\left((\hat{w} - w)H(\hat{w} - w)^T\right) \le \frac{\mu^2 m}{n(2^b - 3)^2} \text{tr}\left(H^{1/2}\right)^2 \|w\|_\infty^2 \left(\log\left(\frac{4mn}{\delta}\right)^2\right).$$

*Proof.* This is a straightforward consequence of the previous lemma. $\square$

**Theorem 15.** *Suppose that we are given an input matrix $w$ with bounded $\|w\|_F$ and we want to quantize it using $b$ bits. Suppose that we first multiply by two-factor orthogonal matrices, and then we re-scale the entries of $w$ by mapping*

$$w_{ij} \mapsto \frac{2^b - 3}{2}\left(\frac{w_{ij}}{\|w\|_F \sqrt{\frac{A^2}{mn} \log\left(\frac{2Cmn}{\delta}\right)^2}} + 1\right) + 1;$$

*this guarantees that $1 \le w_{ij} \le 2^b - 2$. Then, suppose we quantize using the procedure described in the previous lemma. Finally, we undo the scaling and multiplication. Then then with probability at least $1 - \delta$, all the quantized weights will be in range (no overflow or need for clipping) and*

$$\text{tr}\left((\hat{w} - w)H(\hat{w} - w)^T\right) \le \frac{A^4}{n^2(2^b - 3)^2} \text{tr}\left(H^{1/2}\right)^2 \|w\|_F^2 \left(\log\left(\frac{12Cmn^2}{\delta}\right)\right)^6$$

$$= \tilde{\mathcal{O}}\left(\frac{1}{n^2 4^b} \text{tr}\left(H^{1/2}\right)^2 \|w\|_F^2\right).$$

*Proof.* It is a straightforward consequence of Lemma 5, that unioning over the three bounds on the infinity norm of $w$, the incoherence of $H$, and the stochastic rounding, with probability at least $1 - 3\delta$,

$$\text{tr}\left((\hat{w} - w)H(\hat{w} - w)^T\right) \le \frac{m}{n(2^b - 3)^2} \text{tr}\left(H^{1/2}\right)^2 \|w\|_F^2 \left(\log\left(\frac{4mn}{\delta}\right)^2\right)$$

$$\cdot A^2 \log\left(\frac{2Cn^2}{\delta}\right)^2 \cdot \frac{A^2}{mn} \log\left(\frac{2Cn}{\delta}\right)^2.$$

Substituting $\delta \mapsto \delta/3$,

$$\text{tr}\left((\hat{w} - w)H(\hat{w} - w)^T\right) \le \frac{1}{n(2^b - 3)^2} \text{tr}\left(H^{1/2}\right)^2 \|w\|_F^2 \left(\log\left(\frac{12mn}{\delta}\right)^2\right)$$

$$\cdot A^2 \log\left(\frac{6Cn^2}{\delta}\right)^2 \cdot \frac{A^2}{n} \log\left(\frac{6Cn}{\delta}\right)^2.$$

And this right side is clearly less than

$$\mathrm{tr}\left((\hat{w} - w)H(\hat{w} - w)^T\right) \le \frac{A^4}{n^2(2^b - 3)^2} \mathrm{tr}\left(H^{1/2}\right)^2 \|w\|_F^2 \left(\log\left(\frac{12Cmn^2}{\delta}\right)\right)^6.$$

This is what we wanted to show. $\qquad\square$

**Theorem 7.** *Suppose that we run Algorithm 5 (Supplement) to quantize a matrix $W \in \mathbb{R}^{m \times n}$ by solving the objective* (7)*. Then there exists an assignment of the algorithm's hyperparameters $c$ and $\rho$ such that with probability at least $1 - \delta$, all the quantized weights will be in range (no overflow or need for clipping) and*

$$\mathrm{tr}\left((\hat{W} - W)H(\hat{W} - W)^T\right) = \tilde{\mathcal{O}}\left(\frac{1}{n^2 4^b} \mathrm{tr}\left(H^{1/2}\right)^2 \|W\|_F^2\right).$$

*Proof.* This follows directly from the previous theorem, which says explicitly what the hyperparameter assignments should be. $\qquad\square$

## References for the Appendix

[1] Elias Frantar, Saleh Ashkboos, Torsten Hoefler, and Dan Alistarh. Optq: Accurate quantization for generative pre-trained transformers. In *International Conference on Learning Representations*, 2023.

[2] Steve Lalley. Lecture notes on measure-theoretic probability 2. `http://galton.uchicago.edu/~lalley/Courses/383/Concentration.pdf`, 2018.

[3] Markus Nagel, Rana Ali Amjad, Mart Van Baalen, Christos Louizos, and Tijmen Blankevoort. Up or down? adaptive rounding for post-training quantization. In *International Conference on Machine Learning*, pages 7197–7206. PMLR, 2020.

