# OpenReview forum: "QuIP: 2-Bit Quantization of Large Language Models With Guarantees"
_NeurIPS.cc/2023/Conference — NeurIPS 2023 spotlight_

### Official Review · Reviewer_Jvy8 · 2023-06-25

**Soundness:** 3 good
**Presentation:** 3 good
**Contribution:** 3 good
**Rating:** 7
**Confidence:** 4

**Summary:**

This work proposes a unified framework for weight quantization with error feedback together with preprocessing and postprocessing transformation that makes the model more quantization-friendly. Authors derive theoretical bounds on the quantization error and investigate the failure cases of OPTQ quantization. The introduced LDLQ method with incoherence processing is evaluated on quantization of LLM for 2,3 and 4 bit quantization.

**Strengths:**

* Paper proposes a unified view on quantization with the objective of minimization of layer-wise MSE loss that involves well-known OPTQ as a particular case.
* Authors derive explicit average and worst-case bounds on quantization error.
* The introduced incoherence processing is well-motivated and supported by quantitative analysis.
* The demonstrated LDLQ failure case example despite being very different from typical cases occurring in practice is still interesting and shows potential limitations of the optimal quantization methods.
* Different variants of LDLQ achieve strong performance on quantization of models from OPT family at various bit widths considered. The most impressive result is that the model attains reasonable perplexity and zero-shot accuracy for 2 bit quantization, which is known to be very challenging and all the competitive methods experience massive performance drop.
* Overall, the work is well-structured and accompanied with thorough theoretical analysis and empirical study.


**Weaknesses:**

* Method is evaluated only on a single model LLM family. To be sure that the method achieves strong performance on LLM quantization in general one should consider at least one more family of LLM.
* Minor. $U^{\prime}$ in formula (4) is not upper unit triangular but rather strictly upper triangular (I guess  $U^{\prime}  + I$ is supposed to be upper unit triangular).
* Minor.  This will result in each of its eigenvalues being a random unit vector. I guess the authors meant eigenvectors instead of eigenvalues.

**Questions:**

How well does LDLQ+QuIP perform on other families of LLM? Would be interesting to evaluate QuIP on LLaMA given the popularity and impressive performance of this model family. Since it is not fully open-sourced one could consider the recent Falcon family (Falcon-7B and Falcon-40B) instead.

---

> ### Author Rebuttal · Authors · 2023-08-10
>
> **Evaluating on more models.**
> Thanks for the feedback; we agree it is important to evaluate our quantization method on additional models. At the suggestion of the reviewers, we conducted additional experiments on LLaMa-2-70b, and share preliminary results below. We will include a fully comprehensive set of LLaMa results in the revision. Encouragingly, the message of our paper remains the same. QuIP is able to successfully quantize down to 2 bits, as evaluated on Lambada and PiQA zeroshot tasks.
>
> | | Lambada Accuracy | PiQA Accuracy |
> | --  | -- | -- |
> | LLaMa-2-70b-chat 2 Bit | 74.02 | 77.48 |
>
> **Minor linear algebra clarification on Eq. (4)**
> Thanks for catching the typo, yes $U'$ should be strictly upper triangular.
>
> **Line 170 eigenvalue vs eigenvector clarification.**
> Thanks for catching the typo, yes we meant to say eigenvectors.

---

> > ### Comment · Reviewer_Jvy8 · 2023-08-10
> > **Response to rebuttal**
> >
> > After reading the responses, I decided to keep the original score.
> >
> > LLaMA-2-chat is a good choice for the method validation, but the results provided in response involve only 2-bit compression for a single model on two tasks from `lm-eval-harness`. Individual tasks are known to be quite noisy and limited, therefore, in order to make any definite conclusions about the LLM performance it would be desirable to consider a larger number of tasks (at least 5-6). In addition, I would recommend comparing the performance against the fp16 baseline, since these metrics, without having a reference point (one can indeed check the number of LLaMA-2 paper), are not illustrative. Anyway, given that 2 bits is a challenging target, a significant drop in performance is expected and having reasonable performance at this point is still a good result.

---

> > > ### Author Response · Authors · 2023-08-16
> > > **Updated results on LLaMa-2**
> > >
> > > We conducted a suite of further experiments quantizing Llama-2: 7b and 13b parameter models, both pretrained and chat finetuned. Overall, we’ve verified that our method QuIP works well on this additional model, and can provide a step function improvement in quantization at 2 bits compared to OPTQ.
> > >
> > > We adapted code from the GPTQ-for-LLaMa repo, which we use to evaluate WikiText and C4 perplexity. We use the lm-evaluation-harness to evaluate downstream zeroshot accuracy on BoolQ, PiQA, WinoGrande, ARC easy, and ARC challenge.
> > >
> > > | Llama-2 Model | Quant Method | Wbits | Wiki     | C4       | BOOLQ  | PIQA   | WinoGrande | ARC-e  | ARC-c  |
> > > | ------------- | ------------ | ----- | -------- | -------- | ------ | ------ | ---------- | ------ | ------ |
> > > | 13b           | Full Precision | 16    | 4.884    | 6.727    | 80.52% | 80.52% | 72.22%     | 77.48% | 49.15% |
> > > |               | QuIP         | 4     | 5.011    | 6.887    | 80.90% | 80.10% | 72.60%     | 77.30% | 49.20% |
> > > |               |              | 3     | 5.340    | 7.343    | 77.40% | 79.10% | 71.10%     | 76.30% | 49.00% |
> > > |               |              | 2     | 10.094   | 13.131   | 63.90% | 69.60% | 57.90%     | 55.80% | 31.50% |
> > > |               | OPTQ         | 4     | 5.203    | 7.060    | 78.80% | 80.40% | 70.00%     | 76.10% | 48.80% |
> > > |               |              | 3     | 6.666    | 8.910    | 74.50% | 76.70% | 69.30%     | 69.50% | 42.49% |
> > > |               |              | 2     | 3086.167 | 406.934  | 40.20% | 48.80% | 48.70%     | 27.00% | 27.90% |
> > > | 13b-chat      | Full Precision | 16    | 6.108    | 8.489    | 81.71% | 79.11% | 71.27%     | 73.74% | 50.26% |
> > > |               | QuIP         | 4     | 6.275    | 8.733    | 80.80% | 78.40% | 71.20%     | 72.50% | 48.60% |
> > > |               |              | 3     | 6.713    | 9.367    | 78.60% | 77.80% | 70.50%     | 71.30% | 47.10% |
> > > |               |              | 2     | 16.046   | 20.034   | 58.30% | 67.50% | 56.90%     | 53.00% | 33.10% |
> > > |               | OPTQ         | 4     | 6.454    | 8.962    | 79.60% | 78.00% | 70.50%     | 72.80% | 49.40% |
> > > |               |              | 3     | 8.393    | 11.399   | 70.80% | 74.40% | 64.70%     | 66.70% | 42.20% |
> > > |               |              | 2     | 3136.833 | 1138.701 | 41.90% | 47.90% | 47.50%     | 25.50% | 29.60% |
> > >
> > > | Llama-2 Model | Quant Method | Wbits | Wiki   | C4       | BOOLQ  | PIQA   | WinoGrande | ARC-e  | ARC-c  |
> > > | ------------- | ------------ | ----- | ------ | -------- | ------ | ------ | ---------- | ------ | ------ |
> > > | 7b            | Full Precision | 16    | 5.472  | 7.263    | 77.77% | 79.11% | 69.06%     | 74.54% | 46.25% |
> > > |               | QuIP         | 4     | 5.940  | 8.010    | 75.87% | 77.26% | 67.88%     | 71.00% | 42.58% |
> > > |               |              | 3     | 6.499  | 8.738    | 74.92% | 76.28% | 67.01%     | 69.15% | 41.47% |
> > > |               |              | 2     | 27.125 | 31.333   | 52.72% | 60.66% | 51.70%     | 39.14% | 26.19% |
> > > |               | OPTQ         | 4     | 6.067  | 7.845    | 76.09% | 78.51% | 67.96%     | 72.52% | 44.03% |
> > > |               |              | 3     | 9.505  | 11.956   | 68.84% | 74.32% | 63.38%     | 62.16% | 37.37% |
> > > |               |              | 2     | NaN    | 1794.547 | 41.31% | 48.31% | 48.46%     | 26.09% | 27.56% |
> > > | 7b-chat       |  Full Precision |  16   | 7.077  | 9.528    | 80.67% | 76.66% | 66.22%     | 69.65% | 44.37% |
> > > |               | QuIP         | 4     | 7.431  | 10.147   | 80.24% | 76.71% | 66.46%     | 67.26% | 42.49% |
> > > |               |              | 3     | 8.090  | 11.052   | 72.48% | 76.33% | 65.35%     | 67.76% | 40.19% |
> > > |               |              | 2     | 66.586 | 61.662   | 50.18% | 57.67% | 50.43%     | 35.27% | 27.90% |
> > > |               | OPTQ         | 4     | 7.791  | 10.965   | 79.02% | 75.68% | 66.54%     | 69.36% | 41.13% |
> > > |               |              | 3     | 11.847 | 17.736   | 64.25% | 70.67% | 62.19%     | 53.62% | 33.62% |
> > > |               |              | 2     | NaN    | NaN      | 45.54% | 50.00% | 49.88%     | 27.57% | 29.10% |

---

> > > > ### Comment · Reviewer_Jvy8 · 2023-08-16
> > > > **Response**
> > > >
> > > > Thanks for the updates.
> > > >
> > > > In my own evaluations, GPTQ performs considerably better for 3,4 bit quantization than the numbers reported in your table (4 bit GPTQ has lower perplexity than the numbers reported). However, for 2 bits I can confirm that the application of GPTQ to Llama breaks the model completely, whereas QuIP quantized models are still usable.
> > > >
> > > > Nevertheless, considering again the overall contribution of the work, I decided to keep the original score.

---

### Official Review · Reviewer_Rfzg · 2023-07-06

**Soundness:** 3 good
**Presentation:** 3 good
**Contribution:** 3 good
**Rating:** 5
**Confidence:** 3

**Summary:**

The work presented in this paper introduces quantization with incoherence processing, which enables better quantization with fewer bits per parameter. The authors provide a theoretical analysis for adaptive rounding methods and present experimental results demonstrating performance of 2-bit quantization..To do this, the paper introduces the LDLQ adaptive rounding method and demonstrates its optimality compared to other rounding methods which specify linear feedback U for hessians, and rounding to integers (see Theorem 1). The authors define incoherence and demonstrate its effectiveness in achieving a theoretically superior asymptotic bound for LDLQ in terms of the spectrum of H. Additionally, the authors propose efficient pre and post incoherence processing techniques to transform W and H matrices, eliminating the need for nxn matrix multiplications.

**Strengths:**

The paper encompasses numerous interesting new concepts, theorems, and their corresponding proofs. The extensive experimentation serves to prove the authors' claims effectively. The writing is solid, making the paper relatively accessible despite the number of theorems and proofs. Proposed incoherence seems to improve baseline methods, for example OPTQ (table 6,7 appendix). The utilization of the incoherence technique has the potential to contribute significantly to achieve usable 2-bit quantization. They provided code, which is always a plus.

**Weaknesses:**

1) One significant drawback of the paper is that despite the aforementioned enhancements, the performance of 30-bit 2-bit quantization, although much better than 2 bit OPTQ, still falls short compared to 13b model 4-bit quantization with OPTQ(and even 4 bit RTN), making it practically unusable. This limitation diminishes the practical usability surrounding the work on 2 bit quantization.

2) The concept of incoherence remains unclear until page 4 of the paper, which is problematic considering it is one of the main focal points. I believe the introduction should include a clear definition and intuitive explanation of what incoherence entails. Furthermore, I found it challenging to establish a connection between the intuition provided in line 22 and the definition presented in line 134.

3) It is possible I missed, but it appears that there is no mention or measurement of inference speed presented in the paper.

**Questions:**

1)I may have overlooked some details, but it appears that the paper only show how making H incoherent reduce loss upper bounds, nothing about W incoherence. What is benefit for making also W  incoherent?
2) In line 169 you write that "to make symmetric matrix incoherent is to conjugate it by uniform random orthogonal matrix". Can you please add citation or explanation why this is true?
3) Again I may have overlooked some details, in 172- 176 you said that the procedure described there makes the matrix H and W incoherent with high probability/ Why is that, and where is the proof?

Suggestions:
1. In line 64, it seems that "OBC" might be a typographical error, and "OBQ" could be the correct term.
2. It would be helpful to include a citation for LDL decomposition.
3.In line 99, it would enhance clarity to present a equation that demonstrates the result obtained when equation (4) is applied in equation (3).
4.In line 217, it appears that "OTPQ" might be a typographical error, and you mean "OPTQ".

**Limitations:**

Yes

---

> ### Author Rebuttal · Authors · 2023-08-10
>
> **What’s the best use of bits.**
> We conducted a thorough analysis of the experimental data submitted in our paper. The reviewer is accurate in their observation that 2 bit quantization with QuIP is not worthwhile in terms of total memory budget. However, QuIP is the first method we are aware of to achieve decent and usable results at 2 bits quantization. Other methods including OPTQ cannot sensibly quantize at 2 bits.
>
> Our results indicate that QuIP at 3 bit quantization may be the best for a fixed bit budget, assuming our results generalize to other models. Please see the attached figure where we plot total memory vs perplexity/accuracy on the full set of language generation and zeroshot tasks we include in our paper submission. Over all 7 settings and all OPT models up to 30b, QuIP at 3 bits (and 4 bits) quantization achieves a better use of memory than OPTQ quantized to 4 bits, and the full precision models.
>
> Previous quantization methods did not work at 2 bits, but our method gives decent and usable results at this quantization level. Our work indicates that quantizing to 3 bits with QuIP is the best use of a fixed bit budget, and paves the way towards making 2 bit inference viable.
>
>
> **Improving incoherence explanations.**
> Great point, we’ll improve our explanations based on your feedback.
>
> In Definition 1, incoherent Hessian matrices are defined as having sufficiently small eigenvectors. Incoherent weight matrices are defined as having sufficiently small entries. The intuitive explanation in Line 22 states “the weights themselves and the directions in which it is important to have good rounding accuracy are not too large in any one coordinate”. We see the definition of incoherent weight matrices ensures not too large weights. Note the Hessian is in the proxy objective $tr( (\hat{W} - W) H (\hat{W} - W)^T )$, and the eigenvectors of H indicate directions in which rounding accuracy is important. Nagel et. al. [1] also give a nice explanation on the derivation of H.
>
> [1] Markus Nagel, Rana Ali Amjad, Mart Van Baalen, Christos Louizos, and Tijmen Blankevoort. Up or down? adaptive rounding for post-training quantization. In International Conference on Machine Learning, pages 7197–7206. PMLR, 2020.
>
> **Evaluating inference speed.**
> This is a valid point. We have developed a non-optimized implementation including a
> 2-bit CUDA kernel and performed initial inference timing results. Based on these results, the additional overhead of the incoherent matrix multiplies do not translate to a significant slowdown. We conducted sequence generation of length 512 on LLaMa-2-70b quantized to 2 bits, with and without our incoherence processing. Our observations so far indicate that loading the weights onto the GPU dominates the inference time, which is not changed by our method. Further optimizations such as tiling the incoherent matrix multiplies are expected to speed up inference. Section 4.1 also demonstrates why the incoherence processing operations are asymptotically non-dominant.
>
> |     | Inference time (s) |
> | -- | ------------------------ |
> | With incoherent matrix multiplies | 416.575 |
> | Without incoherent matrix multiplies | 377.166 |
>
> Experiments conducted on an NVIDIA RTX A6000 GPU.
>
> **What does our theory say about the benefit of making W incoherent?**
> Theorem 7 shows that the method in Algorithm 4 is theoretically optimal when rounding to a finite grid. Theorem 7 requires that the weights are incoherent, and ensures this by applying Lemma 5 to make the weights incoherent with high probability. The theorem statement could also be rewritten to require weight incoherence as an assumption.
>
> Beyond our theoretical analysis, weight incoherence provides several empirical benefits. Weight incoherence ensures that every weight entry is essentially the same. Therefore the need for different scale factors per group is reduced.
>
>
> **Line 169 question on incoherence.**
> Conjugating a matrix $H$ by a matrix $U$ results in producing the matrix $U H U^{-1}$, which in the case of orthogonal $U$ is equivalent to $U H U^T$. Let $Q D Q^T$ be the eigendecomposition of $H$. It follows that $(UQ) D (UQ)^T$ is an eigendecomposition of $U H U^T$. If $U$ is drawn uniformly at random from the set of orthogonal matrices, then because $Q$ is also an orthogonal matrix, $U Q$ is also a random variable distributed uniformly on the set of orthogonal matrices. It follows that each column of $U Q$ (aka the eigenvectors of $U H U^T$) is a random variable distributed uniformly on the unit sphere. The entries of such random vector in $n$ dimensions concentrate in magnitude $n^{-1/2}$, and so must the entries of $U Q$. Therefore, $U H U^T$ will (with high probability) be incoherent with a small factor $\mu$, and we can get bounds on this factor $\mu$ using concentration inequalities on the magnitude of entries of a random point on the unit sphere).
>
> We will make sure to improve the explanation in the paper.
>
> **Lines 172-176 incoherence with high probability.**
> Please see section 4.1 and Lemma 5 for explanation on how we can ensure the Hessian and weight matrices are incoherent with high probability.
>
> **OBC vs OBQ.**
> Sections 3 and 4 of the OPTQ paper reference the “OBQ” method that is proposed within the “OBC” paper.
>
> **LDL citation.**
> Thanks, we will include a citation for the LDL decomposition.
>
> **Line 99 better clarity when showing Eq.(4) applied to Eq.(3).**
> This is a good point, we will include the additional explanation.
>
> **Line 217 typo.**
> Thanks for pointing this typo out, we will fix it.

---

> > ### Comment · Reviewer_Rfzg · 2023-08-17
> >
> > Sorry for the delay in the response.Thank you for taking time to address my questions. I have read the feedback from other reviewers and the authors' rebuttal.
> >
> > Regarding Weakness 1: Thank you for pointing out results for 3-bit quantization. I understand the fact that 2-bit quantization is challenging, and making it work somewhat decently is achievement by itself. However, from a practical standpoint, despite the fact that QuIP outperforms competitive methods at 2-bit, it is still impractical since for a given memory budget it is better to take smaller model with higher bit-width. Taking into account, the paper focus on 2-bit quantization, in my opinion this is still major drawback. I think one way to address this, is to make paper more transparent, for example, by including tables with fixed memory budget to compare different options.
> >
> > Inference speed: Thank you for measuring the inference speed. I read your discussion with Av5n. I would recommend to include to the paper more precise measurements and fair comparison with OPTQ.
> >
> > Apart from these points, I am satisfied with the authors' response. After revisiting the paper, reading other authors' reviews, and the authors' responses, I would like to increase my score from 4 to 5.

---

### Official Review · Reviewer_Av5n · 2023-07-06

**Soundness:** 3 good
**Presentation:** 3 good
**Contribution:** 3 good
**Rating:** 7
**Confidence:** 4

**Summary:**

- The authors coin a family of adaptive rounding methods for layer-wise quantization, establish that the state-of-the-art method GPTQ is optimal within this family, and prove quality guarantees.
- The paper further introduces incoherence preprocessing, together with a Kronecker-factor based inference scheme, which leads to significant accuracy improvements for very low bitwidth compression.

**Strengths:**

- The paper introduces a non-obvious and useful optimization to GPTQ which fully eliminates the matrix inversion while obtaining equivalent results.
- The authors present what seem like the first theoretical guarantees for an adaptive layerwise rounding algorithm.
- The proposed incoherence preprocessing leads to greatly improved results for 2-bit compression. While the idea of multiplying with an orthogonal matrix to produce more uniform and thus easier to compress data is not new, I have not seen it applied in the context of LLM quantization before.
- The paper also studies the impact of some additional heuristics, like greedy post-processing passes on top of GPTQ. While those are mostly small tricks, it is good to have those implemented and evaluated.
- Code is provided for reproducabilty, including also a script to verify the equivalence of their GPTQ optimization.

**Weaknesses:**

- The method is only evaluated on OPT, which is not considered a great LLM anymore by today's standards; good results on e.g. LLaMa would be significantly stronger.
- There do not seem to be any runtime numbers for inference via the proposed efficient Kronecker-factored scheme. I think that the tripling of required FLOPs (2 additional o(n^2) matmuls if I understood this part correctly) may be challenging to implement in practice without significant overheads (both for low- and large-batch inference), which would limit the practicality of incoherence preprocessing.
- The algorithm family seems to be somewhat designed around how GPTQ works and the corresponding optimality is thus not exactly super surprising.

**Questions:**

- Given that the largest models are usually more robust to quantization, I am wondering how QuiP performs on the 66B and 175B variants, is there any reason why those results are missing?
- How does QuiP perform with groups, which are generally very effective for standard GPTQ / RTN?
- 221: GPTQ only requires one inverse + one Cholesky decomposition
- 282: It would be good to reference that the official GPTQ repository also proposed a similar trick in the context of LLaMa models several months ago.
- The incoherence post-processing code currently returns a matrix that is not quantized anymore, which is a bit confusing. Related to that, I would also suggest to provide pseudocode for the efficient Kronecker-factor inference scheme.

**Limitations:**

The paper briefly discusses limitations and broader impact in the Appendix.

---

> ### Author Rebuttal · Authors · 2023-08-10
>
> **Evaluating on more models.**
> Thanks for the feedback; we agree it is important to evaluate our quantization method on additional models. At the suggestion of the reviewers, we conducted additional experiments on LLaMa-2-70b, and share preliminary results below. We will include a fully comprehensive set of LLaMa results in the revision. Encouragingly, the message of our paper remains the same. QuIP is able to successfully quantize down to 2 bits, as evaluated on Lambada and PiQA zeroshot tasks.
>
> | | Lambada Accuracy | PiQA Accuracy |
> | --  | -- | -- |
> | LLaMa-2-70b-chat 2 Bit | 74.02 | 77.48 |
>
> **Evaluating inference speed.**
> This is a valid point. We have developed a non-optimized implementation including a
> 2-bit CUDA kernel and performed initial inference timing results. Based on these results, the additional overhead of the incoherent matrix multiplies do not translate to a significant slowdown. We conducted sequence generation of length 512 on LLaMa-2-70b quantized to 2 bits, with and without our incoherence processing. Our observations so far indicate that loading the weights onto the GPU dominates the inference time, which is not changed by our method. Further optimizations such as tiling the incoherent matrix multiplies are expected to speed up inference. Section 4.1 also demonstrates why the incoherence processing operations are asymptotically non-dominant.
>
> |     | Inference time (s) |
> | -- | ------------------------ |
> | With incoherent matrix multiplies | 416.575 |
> | Without incoherent matrix multiplies | 377.166 |
>
> Experiments conducted on an NVIDIA RTX A6000 GPU.
>
> **The algorithm family is somewhat designed around GPTQ, and therefore its optimality is not super surprising.**
> We’d like to respectfully push back on this statement. While we believe the LDLQ algorithm formulation and its subsequent in-class optimality proof are intuitive to understand, the equivalence of LDLQ and GPTQ is not intuitive. The GPTQ paper clearly demonstrated its empirical success, but deriving a theoretical proof of optimality based on its method formulation was not clear. Only by showing equivalence to LDLQ could we leverage LDLQ’s optimality to show GPTQ’s in-class optimality, but from looking at their respective formulations this connection is not clear.
>
> **How does QuIP perform on larger OPT models 66B/176B? Why are these missing?**
> Due to time and compute constraints we were unable to submit experiments on larger OPT models. We include some results of our method QuIP on OPT-66b evaluated on language generation tasks, and will include a full set of results on 66B and 176B in the revised paper. Our story on these larger OPT models remains the same: with our incoherence processing we are able to achieve good quantization down to 2 bits.
>
> QuIP on OPT-66b:
> | W Bits | Wiki | PTB | C4 |
> | -------- | ------ | ------ | ---- |
> | 16      | 9.34 | 13.36 | 10.99 |
> | 4        | 9.35 | 13.39 | 11.03 |
> | 3        | 9.45 | 13.55 | 11.17 |
> | 2        | 10.64 | 15.68 | 12.67 |
>
>
> **How does QuIP perform with groups?**
> We tried groupsize 32 and 128 on the OPT-350m model. At 2 bits, we see there is a modest improvement from using grouping. There are interesting research questions here regarding the effect of groupsize with our method. Theoretically, we would not expect groupsize to help too much since incoherence processing should be making the weights relatively uniform. We do however see some benefit at 2 bits for OPT-350m, and further exploration is warranted on additional models.
>
> QuIP on OPT-350m with various groupsizes:
> | quant method  |     | wiki   | ptb    | c4     |
> | -------- | --- | ------ | ------ | ------ |
> | full     | W16 | 22.00  | 31.07  | 22.59  |
> | quip     | W4  | 22.50  | 32.57  | 23.23  |
> |          | W3  | 25.19  | 35.65  | 25.48  |
> |          | W2  | 672.29 | 744.18 | 320.04 |
> | quip-groupsize 32  | W4  | 23.05  | 32.57  | 23.39  |
> |          | W3  | 25.70  | 36.98  | 25.91  |
> |          | W2  | 197.02 | 244.10 | 133.76 |
> | quip-groupsize 128 | W4  | 23.03  | 32.77  | 23.39  |
> |          | W3  | 25.59  | 36.82  | 25.81  |
> |          | W2  | 192.59 | 220.44 | 128.27 |
>
>
> **Clarify number of matrix inverse / Cholesky in GPTQ.**
> We realize Algorithm 1 in GPTQ’s paper states only one inverse + one Cholesky decomposition. However, in [GPTQ’s code](https://github.com/IST-DASLab/gptq/blob/2d65066eeb06a5c9ff5184d8cebdf33662c67faf/gptq.py#L101) we can see two calls to `torch.linalg.cholesky(H)`, and one call to `torch.cholesky_inverse(H)`.
>
> **Acknowledge reordering also used in GPTQ.**
> Thanks for pointing that out, we will make sure to reference the GPTQ repo.
>
> **Clarify incoherence post-processing, in converting back to float. Suggest psuedocode for Kronecker-factor inference.**
> To perform a linear layer forward pass in our current implementation, we (given input activations):
>
>
> (1) perform a multiplication by each of the input-side Kronecker factor matrices (2 small matrix-matrix multiplies)
>
> (2) multiply the result by the quantized weights using a specialized kernel that does quantized-matrix-vector multiplies without having to decompress (as in GPTQ)
>
> (3) multiply the result by each of the output-side Kronecker factor matrices.
>
> This gives a speedup relative to a baseline of offloading the original 16-bit weight matrix to the CPU and loading it to the GPU on-demand, because loading the weights onto the GPU is very expensive due to limited host-device bandwidth. The additional Kronecker factor matrix multiplies slow us down a bit relative to GPTQ, however the slowdown is small since the work of the extra matrix multiplies is asymptotically small.
>
> We will make sure to include psuedocode and an improved explanation in the paper revision.

---

> > ### Comment · Reviewer_Av5n · 2023-08-16
> > **Post-Rebuttal Comments**
> >
> > Thank you for the detailed reply!
> >
> > I am not entirely convinced by your new experimental results, in particular the kernel numbers. 377s for 512 tokens implies ~736ms per token, GPTQ reports 589ms on the same GPU for OPT-175B, a 2x larger model, at FP16. An ideal implementation of a 2-bit kernel for the 2x smaller LLaMa-70B should be close to 16x faster than this (as this is the difference in memory that needs to be loaded, which dominates inference for generation). Hence, your low overheads currently seem to be measured relative to a rather uncompetitive baseline.
> >
> > As for the two Cholesky decompositions,  I believe `torch.linalg.cholesky(H)` and `torch.cholesky_inverse(H)` combine to a symmetric matrix inverse that should be faster and more stable than a general inversion.
> >
> > Nevertheless, I will maintain my score based on the merits of the work discussed in my initial review.

---

> > > ### Author Response · Authors · 2023-08-17
> > > **Clarification on timing experiments**
> > >
> > > Thanks for your response. We were not precise enough in our original rebuttal; our timing results were to generate a batchsize of 8, with sequences length 512 each. Therefore our baseline model had an inference throughput of 92ms/token, not 736ms/token. While this is faster than GPTQ's 589ms/token for OPT-175B at fp16 precision, we've identified several factors that make a direct comparison difficult.
> > >
> > > (1) Our code measures the whole huggingface token generation loop including encoding and decoding, while GPTQ's code just measures the model evaluation. (2) We used 1 GPU, while GPTQ's timing results use 8 GPUs. (3) We used a 2-bit compressed version of Llama-2-70b-chat, while they are using OPT-175B. (4) We used a batch size of 8 while the GPTQ's results used a batch size of 1.

---

### Official Review · Reviewer_t6VC · 2023-07-07

**Soundness:** 3 good
**Presentation:** 3 good
**Contribution:** 4 excellent
**Rating:** 7
**Confidence:** 3

**Summary:**

This paper introduces QuIP, an algorithm for weight quantization in large language models, with theoretical guarantees. The experimental results demonstrate that QuIP achieves nearly lossless performance when using 3-bit quantization for models larger than 3B, and it shows good performance with 2-bit quantization for models larger than 7B.

**Strengths:**

First of all, QuIP is the first post-training quantization algorithm that compresses weights to 2 bits with reasonable loss of performance, significantly pushing the boundary of LLM on-device deployment and making larger LLMs available to ordinary users.

Second, the paper proposes LDLQ rounding method, which is both worst-case and average-case optimal in a family of adaptive rounding methods, i.e. iterative quantization through each column of weights. The family is carefully selected so that OPTQ (previously known as GPTQ) falls within this family.

Third, solid proof is shown that LDLQ is optimal in its family assuming rounding to integers, while a rigid study, with counterexamples and more careful analysis is conducted in more realistic cases in Section 5.2.

**Weaknesses:**

The experiments are mostly conducted on OPT, which might not be the most standard model at the time of reviewing. Models like Llama, Falcon and MPT might be more popular LLMs. However, the reviewer understand the paper was submitted months ago and there might be limited time to conduct all the experiments. Also, the main experiments are conducted on decoder-only auto-regressive language models, while there are other models/domains of interest, for example, FastChat-T5 which is encoder-decoder architecture, Vision transformers, etc.

**Questions:**

As mentioned in Section 5.1., OPTQ is equivalent to LDLQ in the class of adaptive rounding methods with linear feedback (proof in Supplement E), and thus is a special case of LDLQ in general. Is there any further case study why OPTQ performs worse in general?

Is there any analysis on quantization outliers under QuIP/LDLQ and how would QuIP handle those cases?

**Limitations:**

We believe that considering activation quantization could lead to further improvements in the future. However, the implementation of QuIP as a weight quantization method has already overcome major obstacles and enabled the deployment of LLMs.

Currently, there is a lack of realistic experiments to assess the performance, in terms of compute speed. Nevertheless, implementing and optimizing 2/3-bit performance on CUDA would likely be a highly challenging task, and it could be worthwhile to explore this further in a separate research paper.

---

> ### Author Rebuttal · Authors · 2023-08-10
>
> **Evaluating on more models.**
> Thanks for the feedback; we agree it is important to evaluate our quantization method on additional models. At the suggestion of the reviewers, we conducted additional experiments on LLaMa-2-70b, and share preliminary results below. We will include a fully comprehensive set of LLaMa results in the revision. Encouragingly, the message of our paper remains the same. QuIP is able to successfully quantize down to 2 bits, as evaluated on Lambada and PiQA zeroshot tasks.
>
> | | Lambada Accuracy | PiQA Accuracy |
> | --  | -- | -- |
> | LLaMa-2-70b-chat 2 Bit | 74.02 | 77.48 |
>
> **Clarification on LDLQ vs OPTQ.**
> To clarify, we show that OPTQ and LDLQ are in fact equivalent methods. We develop a general class of quantization methods (adaptive rounding with linear feedback), which encompasses LDLQ, OPTQ, as well as nearest and stochastic rounding. Within this general class of methods, we show that LDLQ/OPTQ is worst and average-case optimal (Theorem 1).
>
> One of the main insights of our paper is the benefit of what we call “incoherence processing”. This development greatly improves the performance of all rounding methods at 2 and 3 bits per weight, even nearest rounding.
>
>
> **Outlier analysis.**
> Intuitively, our incoherence processing reduces the prevalence of outliers. When the weight and proxy Hessian matrices are incoherent, the weights themselves and the directions in which it’s important to have good rounding accuracy are not too large in any coordinate. The empirical success of our method indicates the benefit of this outlier reduction.
>
> Theoretically, Definition 1 gives a definition for incoherent weight matrices such that each entry is not too large. Under this incoherence assumption, we show in Lemma 3 that LDLQ can be superior to nearest and/or stochastic rounding, depending on worst or average case settings. Section 5.1 and Lemma 5 show how we can achieve incoherence in practice with high probability, using Kronecker products of random orthogonal matrices.
>
> **Considering activation quantization.**
> We agree that investigating activation quantization is an exciting direction, albeit out of scope for this paper.
>
> **Evaluating inference speed.**
> This is a valid point. We have developed a non-optimized implementation including a
> 2-bit CUDA kernel and performed initial inference timing results. Based on these results, the additional overhead of the incoherent matrix multiplies do not translate to a significant slowdown. We conducted sequence generation of length 512 on LLaMa-2-70b quantized to 2 bits, with and without our incoherence processing. Our observations so far indicate that loading the weights onto the GPU dominates the inference time, which is not changed by our method. Further optimizations such as tiling the incoherent matrix multiplies are expected to speed up inference. Section 4.1 also demonstrates why the incoherence processing operations are asymptotically non-dominant.
>
> |     | Inference time (s) |
> | -- | ------------------------ |
> | With incoherent matrix multiplies | 416.575 |
> | Without incoherent matrix multiplies | 377.166 |
>
> Experiments conducted on an NVIDIA RTX A6000 GPU.

---

### Author Rebuttal · Authors · 2023-08-10

We thank the reviewers for their helpful feedback.

In summary, reviewers noted how the proposed method pushed the boundary of LLM quantization down to 2 bits, provided a novel theoretical understanding of adaptive layerwise rounding algorithms, and conducted extensive experiments. Previous quantization methods did not work at 2 bits, but our method gives decent and usable results at this quantization level. Our work indicates that quantizing to 3 bits with QuIP is the best use of a fixed bit budget, and paves the way towards making 2 bit inference viable.

The primary concerns were regarding:

**(1) Evaluation on additional models:** QuIP achieves good 2 bit quantization on LLaMa-2. We conducted preliminary experiments quantizing another model, LLaMa-2-70b, to 2 bits, and see that it achieves good zeroshot accuracy on Lambada and PiQA tasks.

**(2) Evaluation of inference speed:** Based on a non-optimized implementation, the additional operations from our incoherence processing translate to only a 10% increase in inference time. Additional optimizations are expected to speed up inference time. Section 4.1 also demonstrates why the incoherence processing operations are asymptotically non-dominant.

**(3) Evaluating the optimal bit quantization level for fixed bit budget:** An analysis of the experimental data from our submission shows that quantization with QuIP at 3 bits achieves the best tradeoff curve for total memory and model performance. QuIP at 3 bits makes use of a fixed memory budget better than OPTQ, and the full precision OPT models. This insight is consistent across all 7 of our language generation and zeroshot tasks, and on all OPT models up to 30b.

Full details can be found in the specific reviewer responses. We will include a comprehensive set of experiments and analysis in the paper revision.

**Figure:** QuIP at 3 bits (and 4 bits) achieves a better use of a fixed bits budget compared to OPTQ and full precision OPT models. Evaluated on 3 language generation tasks, 4 zeroshot tasks, and OPT models up to 30b parameters.

---

> ### Author Response · Authors · 2023-08-16
> **Updated results on LLaMa-2**
>
> We conducted a suite of further experiments quantizing Llama-2: 7b and 13b parameter models, both pretrained and chat finetuned. Overall, we’ve verified that our method QuIP works well on this additional model, and can provide a step function improvement in quantization at 2 bits compared to OPTQ.
>
> We adapted code from the GPTQ-for-LLaMa repo, which we use to evaluate WikiText and C4 perplexity. We use lm-evaluation-harness to evaluate downstream zeroshot accuracy on BoolQ, PiQA, WinoGrande, ARC easy, and ARC challenge.
>
> | Llama-2 Model | Quant Method | Wbits | Wiki     | C4       | BOOLQ  | PIQA   | WinoGrande | ARC-e  | ARC-c  |
> | ------------- | ------------ | ----- | -------- | -------- | ------ | ------ | ---------- | ------ | ------ |
> | 13b           | Full Precision | 16    | 4.884    | 6.727    | 80.52% | 80.52% | 72.22%     | 77.48% | 49.15% |
> |               | QuIP         | 4     | 5.011    | 6.887    | 80.90% | 80.10% | 72.60%     | 77.30% | 49.20% |
> |               |              | 3     | 5.340    | 7.343    | 77.40% | 79.10% | 71.10%     | 76.30% | 49.00% |
> |               |              | 2     | 10.094   | 13.131   | 63.90% | 69.60% | 57.90%     | 55.80% | 31.50% |
> |               | OPTQ         | 4     | 5.203    | 7.060    | 78.80% | 80.40% | 70.00%     | 76.10% | 48.80% |
> |               |              | 3     | 6.666    | 8.910    | 74.50% | 76.70% | 69.30%     | 69.50% | 42.49% |
> |               |              | 2     | 3086.167 | 406.934  | 40.20% | 48.80% | 48.70%     | 27.00% | 27.90% |
> | 13b-chat      | Full Precision | 16    | 6.108    | 8.489    | 81.71% | 79.11% | 71.27%     | 73.74% | 50.26% |
> |               | QuIP         | 4     | 6.275    | 8.733    | 80.80% | 78.40% | 71.20%     | 72.50% | 48.60% |
> |               |              | 3     | 6.713    | 9.367    | 78.60% | 77.80% | 70.50%     | 71.30% | 47.10% |
> |               |              | 2     | 16.046   | 20.034   | 58.30% | 67.50% | 56.90%     | 53.00% | 33.10% |
> |               | OPTQ         | 4     | 6.454    | 8.962    | 79.60% | 78.00% | 70.50%     | 72.80% | 49.40% |
> |               |              | 3     | 8.393    | 11.399   | 70.80% | 74.40% | 64.70%     | 66.70% | 42.20% |
> |               |              | 2     | 3136.833 | 1138.701 | 41.90% | 47.90% | 47.50%     | 25.50% | 29.60% |
>
> | Llama-2 Model | Quant Method | Wbits | Wiki   | C4       | BOOLQ  | PIQA   | WinoGrande | ARC-e  | ARC-c  |
> | ------------- | ------------ | ----- | ------ | -------- | ------ | ------ | ---------- | ------ | ------ |
> | 7b            | Full Precision | 16    | 5.472  | 7.263    | 77.77% | 79.11% | 69.06%     | 74.54% | 46.25% |
> |               | QuIP         | 4     | 5.940  | 8.010    | 75.87% | 77.26% | 67.88%     | 71.00% | 42.58% |
> |               |              | 3     | 6.499  | 8.738    | 74.92% | 76.28% | 67.01%     | 69.15% | 41.47% |
> |               |              | 2     | 27.125 | 31.333   | 52.72% | 60.66% | 51.70%     | 39.14% | 26.19% |
> |               | OPTQ         | 4     | 6.067  | 7.845    | 76.09% | 78.51% | 67.96%     | 72.52% | 44.03% |
> |               |              | 3     | 9.505  | 11.956   | 68.84% | 74.32% | 63.38%     | 62.16% | 37.37% |
> |               |              | 2     | NaN    | 1794.547 | 41.31% | 48.31% | 48.46%     | 26.09% | 27.56% |
> | 7b-chat       |  Full Precision |  16   | 7.077  | 9.528    | 80.67% | 76.66% | 66.22%     | 69.65% | 44.37% |
> |               | QuIP         | 4     | 7.431  | 10.147   | 80.24% | 76.71% | 66.46%     | 67.26% | 42.49% |
> |               |              | 3     | 8.090  | 11.052   | 72.48% | 76.33% | 65.35%     | 67.76% | 40.19% |
> |               |              | 2     | 66.586 | 61.662   | 50.18% | 57.67% | 50.43%     | 35.27% | 27.90% |
> |               | OPTQ         | 4     | 7.791  | 10.965   | 79.02% | 75.68% | 66.54%     | 69.36% | 41.13% |
> |               |              | 3     | 11.847 | 17.736   | 64.25% | 70.67% | 62.19%     | 53.62% | 33.62% |
> |               |              | 2     | NaN    | NaN      | 45.54% | 50.00% | 49.88%     | 27.57% | 29.10% |

---

### Decision · Program_Chairs · 2023-09-21

**Decision:**

Accept (spotlight)

**Comment:**

The submission proposes a new and interesting formulation of GPTQ-like quantization methods specifically focused on LLM compression. It derives a new approach (QuIP) which seems to improve the behavior of known approaches specifically in the extreme compression (2bit) range by leveraging incoherence. While the practical results are not yet quite fully-baked (a fact made evident by the interesting discussion), the reviewers and AC agreed that the submission covers a very broad area, and brings a solid set of contributions.

Therefore, I propose to accept.